# SeeA*: Efficient Exploration-Enhanced A* Search by Selective Sampling

**Dengwei Zhao[1], Shikui Tu[1]\*, Lei Xu[1,2]\***

[1]Department of Computer Science and Engineering, Shanghai Jiao Tong University
[2]Guangdong Institute of Intelligence Science and Technology
`{zdwccc, tushikui, leixu}@sjtu.edu.cn`

## Abstract

Monte-Carlo tree search (MCTS) and reinforcement learning contributed crucially to the success of AlphaGo and AlphaZero, and A* is a tree search algorithm among the most well-known ones in the classical AI literature. MCTS and A* both perform heuristic search and are mutually beneficial. Efforts have been made to the renaissance of A* from three possible aspects, two of which have been confirmed by studies in recent years, while the third is about the OPEN list that consists of open nodes of A* search, but still lacks deep investigation. This paper aims at the third, i.e., developing the Sampling-exploration enhanced A* (SeeA*) search by constructing a dynamic subset of OPEN through a selective sampling process, such that the node with the best heuristic value in this subset instead of in the OPEN is expanded. Nodes with the best heuristic values in OPEN are most probably picked into this subset, but sometimes may not be included, which enables SeeA* to explore other promising branches. Three sampling techniques are presented for comparative investigations. Moreover, under the assumption about the distribution of prediction errors, we have theoretically shown the superior efficiency of SeeA* over A* search, particularly when the accuracy of the guiding heuristic function is insufficient. Experimental results on retrosynthetic planning in organic chemistry, logic synthesis in integrated circuit design, and the classical Sokoban game empirically demonstrate the efficiency of SeeA*, in comparison with the state-of-the-art heuristic search algorithms.

## 1 Introduction

In recent years, combining heuristic search algorithms with deep neural networks has demonstrated remarkable performance across a wide range of practical applications, such as board games [48, 50, 49, 66], video games [46, 64, 67], traveling salesman problem [8, 59], *de novo* drug design [42], retrosynthetic planning [6, 47, 68], logic synthesis [9], and so on. The search algorithm is a slow reasoning process, and heuristic functions serve as counselors to narrow down the search space [2]. Therefore, the effectiveness of search algorithms is significantly influenced by the quality of the guiding functions.

Monte-Carlo tree search (MCTS) is a widely-used, effective algorithm for combinatorial problems. However, if the backup value in MCTS is provided by a heuristic estimator rather than actual rewards, the convergence to the true state value is not guaranteed, leading to compromised search performance. In single-agent problems such as combinatorial puzzles, neural-guided MCTS tends to have a relatively long runtime and often generates solutions that are considerably longer than the shortest path [1].

---

\*Correspondence authors are Shikui Tu and Lei Xu.

A* search [26] is a best-first search algorithm that expands nodes with the minimum total path value $f$ at each step. The evaluation function $f(n)$ on a node $n$ is defined as the summation of $g(n)$, the accumulated cost from the initial node $n_0$ to $n$, and $h(n)$, the expected cost from $n$ to the goal, i.e.,

$$f(n) = g(n) + h(n). \tag{1}$$

Notice that $g(n)$ computes the cost from the known searching trajectory, whereas $h(n)$ is a heuristic function to estimate the cost of the future path from $n$ to the goal. In practice, it is usually difficult to obtain an accurate $h(n)$. A* search is guaranteed to find the optimal solution if $h(n)$ is admissible, i.e., $h(n)$ never overestimates the real cost to the goal. However, due to its best-first expansion strategy, A* has limited exploration capability. If $f(n)$ deviates from the true cost function $f^*(n)$ too much, A* search may become trapped in local optimal branches, and significant efforts are required to resume expansion within the optimal branch. Consequently, the computational efficiency of A* search is compromised in practical applications, even though the optimality of A* might still hold under the guidance of $f(n)$.

MCTS and A* both perform heuristic search. MCTS and reinforcement learning with the help of deep learning contributed crucially to the successes of AlphaGo and AlphaZero, which aroused the interest of comparing MCTS and A* for possible mutual benefits. Deep learning is also able to contribute to the renaissance of A*, three possible aspects are addressed with a family of possible improvements proposed under the name of Deep IA-search [61]. The first and also straightforward aspect is estimating $f(n)$ with the help of deep learning, which makes current studies on A* including this paper into the era of learning aided A*. The second aspect is seeking a better estimation of $f(n)$ with the help of global or future information, featured by two typical mechanisms. One is lookahead or scouting before expanding the current node to collect future information to revise $f(n)$ of the current node, which takes a crucial rule for the success of AlphaGo [48] and also used more than 30 years ago in Algorithm CNneim-A [62]. The other is path consistency, that is, $f(n)$ values on one optimal path should be identical, which has been further confirmed in recent studies [66, 67, 68]. This third aspect is about selecting nodes among the OPEN list that consists of open nodes of A*. It is an old tune even in the classical era of A*, e.g., one suggestion is dividing OPEN into two sublists OPEN and WAIT according to a priori and a posteriori in a Bayesian evaluation [61]. However, investigation is seldom made on what are effective and efficient ways for selecting among OPEN.

In this paper, SeeA* search (short for **S**ampling-**e**xploration **e**nhanced **A***) algorithm is proposed by incorporating exploration behavior into A* search to target at the third aspect. The main contributions are summarized below.[2]

- SeeA* search employs a selective sampling process to screen a dynamic candidate subset $\mathcal{D}$ from the set $\mathcal{O}$ of open nodes that are awaiting expansion. The next expanding node is selected from $\mathcal{D}$, and it may not be the node that has the best heuristic value in $\mathcal{O}$ and will be selected by A*, enabling SeeA* to explore other promising branches. To reduce the excessive expansion of unnecessary nodes during exploration, only the candidate node with the best heuristic value is expanded. Three sampling strategies are introduced to strike a balance between exploitation and exploration. The search efficiency is improved especially when the guiding heuristic function is not accurate enough.

- We theoretically prove that SeeA* has superior efficiency over A* search when the heuristic value function deviates substantially from the true state value function. SeeA* achieves a reduced number of node expansions to identify the optimal path. This performance improvement becomes more pronounced as the complexity of the problems increases and the reliability of the guiding heuristics decreases.

- Experiments are conducted on two real-world applications, i.e., the retrosynthetic planning problem in organic chemistry and the logic synthesis problem in integrated circuit design, as well as the classical Sokoban game. SeeA* outperforms the state-of-the-art heuristic search algorithms in terms of the problem-solving success rate and solution quality while maintaining a low level of node expansions.

---

[2]The source code is available at `https://github.com/CMACH508/SEEA_star`.

## 2 Related work

MCTS [5, 13] utilizes random sampling and tree-based search to efficiently explore search space. Upper Confidence bounds applied to Trees with predictor (PUCT) have been employed by AlphaZero [49], achieving super-human performance in board games. A* search is widely employed for solving optimization problems, such as route planning [54, 53], cubic and puzzle games [1], robotics [17], and so on. Many variants of A* search have been proposed for performance improvement. Weighted A* search (WA*) [18] biased the expanding policy towards states closer to the goal by

$$n^* = \arg\min_n g(n) + \varepsilon h(n), \tag{2}$$

where $\varepsilon$ is a hyperparameter to adjust the weight of the heuristic estimation $h$. WA* with iteratively decreasing weights is employed by the LAMA planner [27, 44], achieving promising results in various domains including Sokoban. DeepCubeA [1] trained heuristic functions by reversing solution pathways from the goal state to guiding the search process of WA*. Commonly, WA* traded optimality for speed, and increasing $\varepsilon$ was assumed to result in faster searches. Additionally, the greedy search based on $h$ values was considered the fastest search. However, empirical observations revealed that increasing $\varepsilon$ slowed down the search in some domains. Greedy search is fast if and only if there is a strong correlation between the heuristic estimations and the true distance-to-go, or if the heuristic is extremely accurate [55]. However, constructing a reliable heuristic function for complicated problems is challenging attributed to the vast search space and the difficulties associated with sample collection in real-world applications. Poor generalization performance also remains a pervasive issue across diverse practical domains, such as retrosynthetic planning. This paper sets out to develop an efficient search algorithm designed to minimize the adverse effects of inaccurate predictions by heuristic functions.

There have been some preliminary studies on the integration of exploration into the A* search. $\varepsilon$−greedy node selection was incorporated into LAMA, suggesting that exploration can improve the coverage of search algorithms even multiple enhancements were already employed [52]. Type-WA* [11] augments WA* with type-based exploration [57] in the focal list [40]. The search space nodes are divided into $T$ distinct groups, and one of these groups is randomly chosen to determine the expanded node. Levin tree search (LevinTS) [38] combined a penalization mechanism based on node depth to encourage exploration for A* search. Policy-guided heuristic search (PHS) [39] generalized LevinTS by introducing a heuristic factor, guided by both a value function and a policy. When the guiding heuristics are sufficiently accurate, the best-first search achieves optimal efficiency without the need for exploration. Insufficient exploration leads the search algorithm to be trapped in local optima guided by inaccurate heuristics. As the accuracy of the guiding heuristic diminishes, the importance of exploration becomes more pronounced in order to mitigate the potential misguidance.

Search algorithms have played a crucial role in solving diverse real-world problems, such as retrosynthetic planning and logic synthesis. Retrosynthetic planning aims to identify a feasible synthetic route using known available building block molecules for a given target molecule. Considering that the synthesis of target molecules typically requires multiple steps and each step encompasses a substantial number of potential chemical reactions, retrosynthetic planning is formulated as a search problem to identify the optimal synthetic pathway. Both MCTS [28, 47, 65] and A* search, such as Retro* [6] and its descendants [24, 30, 33, 58], have demonstrated promising results in retrosynthetic planning. Logic synthesis (LS) is a crucial step in the design of integrated circuits, mapping the high-level logic circuit description into gate-level implementation. In recent years, reinforcement learning algorithms [10, 29, 34, 41, 69] and search methods [9, 37] have shown promising results in the field of LS. Besides, Sokoban is an NP-hard [16] and PSPACE-complete [14] problem, which is a benchmark problem for evaluating the performance of artificial intelligence planning algorithms. Recently, combining reinforcement learning algorithms with search-based methods has demonstrated remarkable performance in effectively solving the Sokoban problem [19, 20, 22, 31, 43].

## 3 Preliminaries and limitations on A* search

Single-agent problems solved in this paper are formulated as Markov decision processes. Let $\mathcal{N}$ represent the set of nodes in the search tree, where each node $n \in \mathcal{N}$ corresponds to a state $s$ in the state space $\mathcal{S}$. The set of $n$'s children is represented as $CH(n)$. The root of the tree and the initial state are denoted as $n_0$ and $s_0$ respectively. At each interactive step, action $a_t \in \mathcal{A}$ is

applied to the current state $s_t$, resulting in the subsequent state $s_{t+1} = \mathcal{T}(s_t, a_t)$ and transition cost $c_{t+1} = c(s_t, a_t)$, where $\mathcal{T}$ is the state transition function to obtain the following state $s_{t+1}$ when taking action $a_t$ at state $s_t$, and $c$ is the cost function gieving the received cost when taking action $a_t$ at state $s_t$..

The search tree of A$^*$ contains two distinct types of nodes: *closed nodes*, which have already been expanded, and *open nodes*, which are waiting to be expanded [26]. Let $\mathcal{O}$ and $\mathcal{C}$ denote the set of open nodes and closed nodes respectively. The search process of A$^*$ can be summarized as follows:

- Step 1: Initialize $n_0$ with $s_0$, and mark it as open node by setting $\mathcal{O} \leftarrow \{n_0\}, \mathcal{C} \leftarrow \emptyset$.
- Step 2: Select the node $n$ with the lowest total path cost $f(n)$ from the open set $\mathcal{O}$, i.e., $n = \arg\min_{n' \in \mathcal{O}} f(n')$.
- Step 3: If the node $n$ is the goal, terminate the search process successfully. Otherwise, expand the node $n$, and update $\mathcal{C} \leftarrow \mathcal{C} \cup \{n\}, \mathcal{O} \leftarrow \mathcal{O} \cup CH(n) \setminus \{n\}$.
- Step 4: Repeat step 2 and 3 until $\mathcal{O}$ becomes empty, or exceeding the predetermined maximum runtime or the number of expanded nodes, terminating with failure.

A$^*$ search always selects the node with the best heuristic value from the open set without exploration. When the heuristic function $f$ can accurately estimate the true cost $f^*$, this best-first search is the most efficient. However, if the estimation by $f$ is not accurate enough, the node with the minimum $f$ value may not correspond to the optimal one, which instead has the lowest $f^*$ value. The search process might be trapped in a local optimal branch, and substantial computational efforts are required to resume expansion on the optimal branch, which diminishes the efficiency of the search algorithm. Considering an example in Figure 1(a), suppose the cost for each step (or edge) on the optimal path is 100, and on the non-optimal path is only 1. The true total path cost at any node $n$ is given by $f^*(n) = g(n) + h^*(n)$, where $g(n)$ is given by adding the costs from the root to the node $n$, and the real future cost $h^*(n)$ is a summation of all costs from $n$ to the end (or terminal state). Suppose the evaluation function $f(n) = g(n) + h(n)$ by Equation 1 is exact on the optimal path but underestimates the real cost otherwise. Specifically, define the heuristic function $h(n)$ as follows:

$$h(n) = \begin{cases} h^*(n), & \text{if } n \text{ is on the optimal path} \\ 0, & \text{Otherwise} \end{cases} \tag{3}$$

Then, $h(n)$ satisfies the admissible assumption as it never overestimates the cost, and $h(n) \leq h^*(n)$ is established for all nodes. Therefore, A$^*$ is guaranteed to find the optimal solution guided by $h(n)$ in Equation 3. However, as illustrated in Figure 1(b), guided by the defined heuristic $h$, the nodes on the optimal path will not be expanded until all nodes on non-optimal branches with depths less than 200 have been expanded. The optimal solution is achieved within two steps under the guidance of $f^*$, and the search efficiency of A$^*$ search is largely compromised when $f(n)$ is not accurate enough.

## 4 Method

SeeA$^*$ search is proposed on the basis of A$^*$ search by introducing a candidate set $\mathcal{D}$ of open nodes to provide exploration behavior. Three selective sampling strategies are presented for constructing the candidate set. Moreover, we present a theoretical analysis on the efficiency of SeeA$^*$.

### 4.1 SeeA$^*$ search algorithm

SeeA$^*$ employs the following two steps to replace the Step 2 in A$^*$ search. First, a selective strategy is employed to sample a set of candidate nodes $\mathcal{D}$ from the opening set $\mathcal{O}$. Then, the node $n$ with the lowest $f$-value from the candidate set $\mathcal{D}$, instead of $\mathcal{O}$, is chosen to be expanded in Step 3. The details of SeeA$^*$ are summarized in Algorithm 1 in Appendix A.

- Step 2$a$: Sample a candidate subset $\mathcal{D}$ from $\mathcal{O}$.
- Step 2$b$: Select the node $n$ with the lowest $f$-value from the candidate set $\mathcal{D}$.

As illustrated in Figure 1(c)&(d), if the node with minimum $f$-value is not sampled into the candidate set $\mathcal{D}$ in Step 2$a$, the node selected to be expanded later is not the same as the one by A$^*$ search, which activates exploration on other branches. Step 2$b$ excludes the unpromising nodes by the $f$-value.

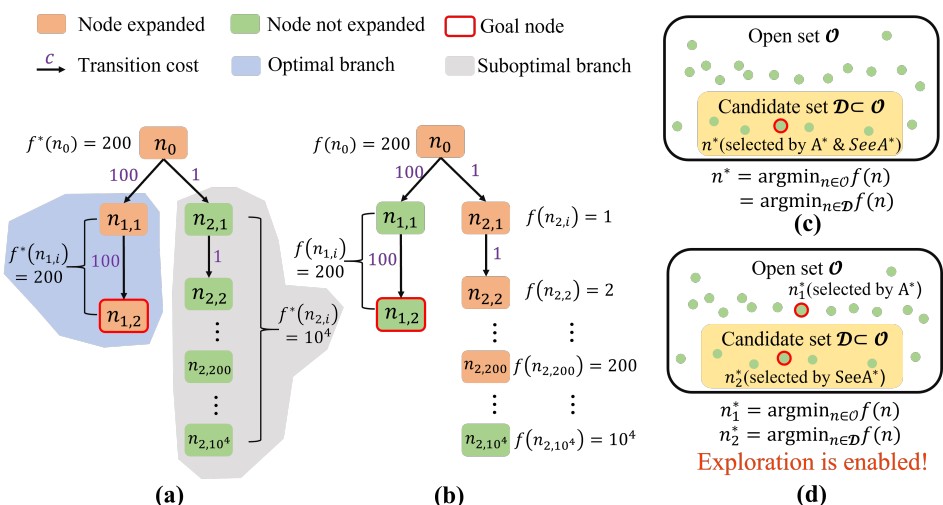

Figure 1: An illustration of how SeeA* overcomes the A* search's limitation. (a) An example of the search tree is guided by the true optimal value $f^*(n) = g(n) + h^*(n)$. Values on the edge denote the cost of each step. (b) On the same example, the A* search is trapped in a suboptimal branch misled by the unreliable heuristics, i.e., $f(n) = g(n) + h(n)$. (d) When the candidate set does not contain the node $n_1^*$ with the best $f$ value, $n_2^*$ will be selected and explored, where $n_2^* \neq n_1^*$.

#### 4.1.1 Uniform sampling strategy

Uniform sampling guarantees an equal selection probability for each node, thereby generating a representative subset that has the same distribution of the population. The procedure is given in Algorithm 2 in Appendix A. If the desired number of candidate nodes, denoted as $K$, is greater than the number of open nodes, the open set $\mathcal{O}$ is used as $\mathcal{D}$. Otherwise, $K$ nodes are randomly selected from the open nodes as $\mathcal{D}$. It should be noted that SeeA* with uniform sampling is different from the $\varepsilon$-Greedy method. The $\varepsilon$-Greedy activates exploration with probability $\varepsilon$ and then uniformly samples a node for expansion, which may expand low-quality nodes. In Step $2a$ of SeeA*, uniform sampling is very likely to include at least one high-quality node with a reasonably low $f$-value and the node will be selected to expand in Step $2b$. More discussions are referred to Appendix P.

#### 4.1.2 Clustering sampling strategy

In the uniform sampling strategy, each node is selected with equal probability. However, there is a non-negligible probability that all sampled nodes are of low quality, leading to the exclusion of nodes along the optimal expansion path from the candidate set $\mathcal{D}$. Therefore, a clustering sampling strategy is proposed, and it partitions open nodes into multiple clusters and subsequently sampling nodes from each cluster, as illustrated in Figure 2 in Appendix B. At least one node from each cluster is sampled compulsorily. Consequently, the probability of including nodes on the optimal branch is increased, thereby facilitating search efficiency. On the other hand, uniform sampling strategy is equivalent to assume that the nodes follow a Gaussian distribution, whereas clustering sampling strategy assumes that the nodes follow a Gaussian mixture distribution from multiple clusters, which provides a more descriptive representation for sampling.

To reduce computational costs, competitive learning [51] is utilized for node clustering. After each node expansion, the incorporation of newly generated nodes into the set $\mathcal{O}$ resembles the process of online sample acquisition in competitive learning. A clustering process is conducted simultaneously with the search process. Offline clustering algorithms, such as K-means or Gaussian mixture model, require recalculating the clustering when incorporating new nodes, thereby imposing additional computational overhead. Each node is represented by a vector extracted by a function $f_h$. $N_c$ cluster centers are randomly initialized as vectors with the same dimension of node embedding. During each expansion, the newly generated nodes are assigned to the cluster with the closest center separately, and the cluster center is updated by moving toward the position of the freshly added node. While preparing the candidate set $\mathcal{D}$, nodes are sampled evenly from each cluster, and uniform sampling is employed to select nodes from each cluster. Details are displayed in Algorithm 3 & 5 in Appendix A.

### 4.1.3 UCT-like sampling strategy

In AlphaZero [49], PUCT achieved a good balance between exploitation and exploration with promising results. In light of this, a UCT-like sampling strategy is proposed. Due to the absence of Monte Carlo simulations, estimated $f$ values are employed to substitute the $Q$ value in PUCT, which is the average backup value obtained from multiple MCTS simulations. The depth of the node is employed as the penalization for exploration [38]. Each node is evaluated by

$$E(n) = f(n) - c_b \times \frac{\sqrt{d_{max}}}{1 + d(n)}, \tag{4}$$

where $c_b$ is an adjustable hyperparameter, $d(n)$ is the depth of node $n$, and $d_{max}$ is the maximum depth of the open nodes. Nodes with smaller $d(n)$ are more likely to be included in the candidate set for exploration. Despite potential errors in $f$ value estimation, it remains a viable node evaluation metric to sample high-quality nodes, and the exploration term is beneficial in mitigating misleading of prediction errors. c to constitute the candidate set $\mathcal{D}$. The details are summarized in Algorithm 4 in Appendix A.

## 4.2 Efficiency of SeeA* search

We further provide a theoretical analysis on the efficiency of SeeA*, demonstrating that SeeA* is superior to A* when the guiding heuristic function $f$ does not estimate the true cost $f^*$ accurately enough. It was claimed in A* search [26] that the $f^*$ values of all nodes on the optimal path are equal to the same cost $\mu_0^f$ and lower than the $f^*$ value of nodes outside the optimal path, which was assumed to be sampled from a Gaussian distribution in [62]. In this paper, the prediction error for $f^*$ is assumed to follow a uniform distribution. Here, Gaussian distribution is denoted as $\mathcal{G}(\cdot, \cdot)$ and uniform distribution is denoted as $\mathcal{U}(\cdot, \cdot)$. Formally, an assumption is made as follows.

**Assumption 4.1** *For each node $n$ on the optimal path, $f(n) \sim \mathcal{U}(\mu_0^f - \sigma, \mu_0^f + \sigma)$. For nodes not on the optimal path, $f(n) \sim \mathcal{U}(f^*(n) - \sigma, f^*(n) + \sigma)$, and $\{f^*(n)\}$ are independently and identically sampled from $\mathcal{G}(\mu_1^f, \sigma_s^2)$.*

The $\mu_0^f$ and $\mu_1^f$ are the expected total cost for optimal and non-optimal solutions, respectively. The inequality $\mu_0^f < \mu_1^f$ holds because the optimal path has a lower cost. The $\sigma$ represents the magnitude of the prediction error, and the $\sigma_s^2$ is a constant as the variance. Under Assumption 4.1, we can derive:

**Corollary 4.2** *For a node $n$ on the optimal path and a node $n'$ off the optimal path, the probability*

$$p_\sigma = P\left(f(n) \leq f(n')|\sigma\right) \tag{5}$$

*decreases as the prediction error $\sigma$ increases.*

It is worth noting that the establishment of Corollary 4.2 is not limited by the assumption of a uniform noise distribution in Assumption 4.1. When the noise follows a Gaussian distribution, Corollary 4.2 is still established. Refer to Appendix C for more detailed derivations.

Without loss of generality, assume the open set $\mathcal{O}$ contains $N_o$ nodes, $\{n_1, n_2, \cdots, n_{N_o}\}$, and $n_1$ is the optimal node. The probability of A* search expanding node $n_1$ is

$$P_A(\sigma) = P\left(n_1 = \arg\min_{n' \in \mathcal{O}} f(n')|\sigma\right) = \prod_{n' \in \mathcal{O} \setminus \{n_1\}} P\left(f(n) \leq f(n')|\sigma\right) = p_\sigma^{N_o - 1}. \tag{6}$$

SeeA* expands $n_1$ with probability

$$P_S(\sigma) = P\left(n_1 \in \mathcal{D}, n_1 = \arg\min_{n' \in \mathcal{D}} f(n')|\sigma\right) = P(n_1 \in \mathcal{D}) \prod_{n' \in \mathcal{D} \setminus \{n_1\}} p_\sigma. \tag{7}$$

If the uniform sampling strategy is used to select $K$ candidates,

$$P_S(\sigma) = \frac{K}{N_o} p_\sigma^{K-1}. \tag{8}$$

Based on Equation 6 & 8, when the prediction error $\sigma$ is large, SeeA* expands the optimal node with a higher probability than A* search at each step, which is given by the following theorem.

**Theorem 4.3** $P_S(\sigma) > P_A(\sigma)$ *holds if and only if*

$$p_\sigma < H(N_o), \quad where \ H(N_o) = \left(\frac{K}{N_o}\right)^{\frac{1}{N_o-K}}, N_o > K \geq 1. \tag{9}$$

$H(N_o)$ is a monotonically increasing function with respect to $N_o$ which is the size of the open set. With increasing branching factors and longer solution paths for more complex problems, $N_o$ grows and $H(N_o, K)$ monotonically increases with respect to $N_o$. Especially, we have

$$\lim_{N_o \to +\infty} H(N_o) = 1. \tag{10}$$

In this situation, Inequality 9 holds. SeeA* tends to demonstrate superior performance compared to A* in solving complex problems.

Notice that if the heuristic function $f$ predicts the true cost $f^*$ without error, it leads to $p_\sigma = 1$ in Equation 5. Then, Equation 9 does not hold, and in this case, A* search becomes more efficient than SeeA*. However, learning an accurate heuristic function for complex real-world problems is quite challenging, and large prediction errors usually exist, which leads to small $p_\sigma$ and the establishment of Equation 9. The number of candidate nodes $K$ is a key hyperparameter to balance the exploitation A* and the exploration introduced by SeeA*. $P_S(\sigma)$ in Equation 8 reaches its maximum value when $K^* = -1/\log p_\sigma$. When $p_\sigma$ approaches 1, $K^*$ will be the largest $\infty$. In this situation, the candidate set is the same as the open set, and SeeA* degenerates into best-first A*. For small $p_\sigma$, the optimal $K^*$ is the smallest value 1 and SeeA* becomes random sampling. An appropriate value of $K$ should be selected according to the specific situation. According to Equation 7, $P_S(\sigma)$ is related to both $p_\sigma$ and $P(n_1 \in \mathcal{D})$. Utilizing more efficient sampling algorithms than uniform sampling is also capable to enhance the performance of SeeA*. The clustering sampling and UCT-like sampling aim to achieve a higher $P(n_1 \in \mathcal{D})$ by constructing a more diverse candidate set, thereby enhancing the likelihood of expanding the optimal node.

For simplicity, suppose the probability of selecting the optimal node in a single expansion is $P$, and the probability for expanding the optimal node becomes $1 - (1 - P)^\tau$ after $\tau$ expansions. To achieve a probability level of $P_{min}$ for expanding the optimal node, we have

$$\tau \geq \frac{\log\{1 - P_{min}\}}{\log\{1 - P\}}. \tag{11}$$

Based on Theorem 4.3 and Equation 11, SeeA* is more efficient than A* search as it requires fewer expansions to find the optimal solution. It is noted that Equation 9 is derived on the uniform sampling strategy. For a more effective sampling strategy with a higher probability $P(n_1 \in \mathcal{D})$, SeeA* will become more efficient as $P_S(\sigma)$ increases.

## 5 Experiments

Real-world problems are usually complicated, and the amount of available samples for training the heuristic functions is typically small.Two real-world applications, i.e., retrosynthetic planning in organic chemistry and logic synthesis in integrated circuit (IC) design, are considered to evaluate the effectiveness of the proposed method. Since the molecular structures have enormous diversity but in contrast the available experimental data are very limited, the heuristic function to estimate the synthesis cost in retrosynthetic planning suffers from noticeable overfitting problems [68]. Furthermore, the vast chemical reaction space gives rise to a substantial number of branching factors in the search tree, leading to a rapid growth in the quantity of open nodes throughout the search process. Logic synthesis is another practical problem where it is challenging to train a reliable heuristic function to evaluate the solution's quality, due to the immense diversity of circuit functionalities and variations in design methodologies. Therefore, the above two real-world problems are suitable benchmarks to verify the efficiency of SeeA* when the heuristic function is not accurate enough. In addition, Sokoban is a widely-used benchmark for combinatorial optimization solvers. It only permits a maximum of four legal actions at each step, and simulations can be leveraged to generate a substantial amount of data for training high-quality heuristic value estimators. Sokoban is included to verify the impact of an accurate heuristic function on the searching performance. All experiments are conducted using NVIDIA Tesla V100 GPUs and an Intel(R) Xeon(R) Gold 6238R CPU.

## 5.1 Results on retrosynthetic planning

Chemical synthetic pathways are transformed into search trees following the literature [47]. A state is a set of molecules that are able to synthesize the target molecule. The initial state contains only the target molecule. The edges in the search tree represent the chemical reactions that enable state transitions between the connected nodes. The retrosynthetic planning problem is solved if all molecules within a state are available building blocks. A single-step retrosynthetic prediction model is utilized as the policy model to generate potential chemical reactions yielding the input molecule. The 50 chemical reaction templates with the highest probabilities constitute the set of valid actions for the current state. A heuristic function is employed to estimate the synthesis cost of the molecule, given the available building blocks. Each molecule is encoded using a 2048-dimensional Morgan Fingerprint vector [45] as the input for the heuristic functions. Both the single-step retrosynthetic prediction model and the cost estimator are provided by Retro$^*$+ [30] and used to guide the search algorithm. Details about the guiding heuristics are in Appendix D. The last hidden layer's output of the cost estimator is employed as the embedding representation of the input molecule.

Experiments are conducted on the widely-used USPTO benchmark, comprising 190 molecules [6]. Commercially available molecules in *eMolecules*[3] are used as building blocks. Since the invocation of the single-step retrosynthetic prediction model contributes the majority of the computational cost, all search algorithms are limited to a maximum of 500 single-step model calls, or 10 minutes of real-time, following previous works [6, 30]. The outputs of the single-step model are cached to avoid duplicate computation when the same molecule is encountered again [36]. The size of the candidate set is set to $K = 50$. In the clustering sampling, the parameter $\eta$ is set to 0.15, and the number of clusters is 5. In the UCT-like sampling, the parameter $c_b$ is set to 0.35. Additional pruning is not considered. Since the prior policy is already clipped at a minimum value of 0.001, Bayes mixing with a uniform policy to avoid zero-probability is not used in LevinTS [38] and PHS [39].

The results on the USPTO benchmark are reported in Table 1[4]. Due to the exploration induced by selective sampling, the three SeeA$^*$ variants achieve superior performance in terms of the percentage of solved molecules and the average solution length while utilizing minimal wall-clock runtime. Among the three sampling strategies, the UCT-like sampling strategy achieves the best balance between exploration and exploitation. As in the literature [47], predicting the synthetic cost of molecules is challenging, and the cost estimator is not accurate with a non-negligible prediction error $\sigma$. Then, it is expected and consistent with Theorem 4.3 that best-first search algorithms, including WA$^*$ and PHS, are less efficient because they excessively rely on the values of the heuristic function. MCTS requires more node expansions for problem-solving and generates solutions with longer lengths, which is consistent with the findings in the resolution of combinatorial puzzles [1]. The $\varepsilon$-Greedy node selection [52] achieves a success rate of 92.11%, surpassing the performance of A$^*$ search and demonstrating the practical benefits of introducing exploration when the reliability of guidance heuristics is compromised.

Six additional datasets are collected from the literature for further comparisons. These datasets comprise 4719 molecules, much more than the USPTO dataset. Details of the datasets are referred to the Appendix E. According to the results in Table 3 & 4 in Appendix G, SeeA$^*$ maintains its superiority over other search algorithms, and SeeA$^*$(Cluster) has the highest mean success rate of 63.56%. The clustering sampling and UCT-like sampling are better than uniform sampling in terms of the solved rate and the route length, indicating that the utilization of a superior sampling strategy is beneficial for the performance of SeeA$^*$.

## 5.2 Results on logic synthesis

For the logic synthesis problem, a Verilog-based hardware design is first converted into an and-inverter-graph (AIG) representation, and then the AIG is optimized to have the lowest area-delay product (ADP) through a sequence of functionality-preserving transformations. The optimization is combinatorial because the sequence is constructed by selecting transformations one-by-one in order from a set. Following the literature, here the set is formed by seven legal transformations, and the sequence length is fixed at 10. The *resyn2* transformation sequence is used as a baseline for comparisons [9, 10, 37]. More details about logic synthesis are in Appendix H. During the search

---

[3]`http://downloads.emolecules.com/free/2023-12-01/`

[4]Red entries indicate the top rank, while blue entries signify the second position. Table 2 remains consistent.

Table 1: Test results on the USPTO benchmark for retrosynthetic planning problem.

| Algorithm | Solved (%) ↑ | Length ↓ | Expansions ↓ | Avg time (in seconds) ↓ |
|---|---|---|---|---|
| Retro* [6] | 86.84 | 9.71 | 157.11 | 110.57 |
| Retro*+ [30] | 91.05 | 8.74 | 100.15 | 61.24 |
| MCTS($c_{puct} = 4.0$) | 89.47 | 8.23 | 122.97 | 87.86 |
| A* search | 88.42 | 9.27 | 92.45 | 96.07 |
| WA* ($\varepsilon = 1.5$) | 84.21 | 10.16 | 106.97 | 120.69 |
| LevinTS [38] | 96.84 | 7.45 | **57.11** | 39.77 |
| PHS [39] | 87.37 | 10.19 | 93.38 | 108.85 |
| $\varepsilon$-Greedy ($\varepsilon$=0.1)[52] | 92.11 | 43.78 | 89.14 | 76.59 |
| SeeA*(Uniform) | 96.84 | 7.34 | 72.08 | 49.77 |
| SeeA*(Cluster) | 98.42 | 6.48 | 79.75 | 37.98 |
| SeeA*(UCT) | **98.95** | **6.36** | 62.07 | **32.38** |

process, the immediate reward is set to be 0, and the reward of the terminal step is the final AIG's ADP reduction rate against the baseline *resyn2*. The ADP score is approximately computed by ABC [4]. A heuristic function is employed to predict the accumulated reward of a sequence when only a front part of the sequence is available as input in the search process. This function serves as a guiding heuristic for the search algorithms. Following ABC-RL [9], the training dataset consists of 15 circuits, while the test dataset comprises 12 MCNC circuits denoted as $\{C_1 \sim C_{12}\}$ [63] (See Appendix I for more details.). The architecture of the heuristic function, the training and test details are referred to Appendix J. WA* is equivalent to A* search because $g = 0$ in Equation 2.

The results on the MCNC benchmark are presented in Table 2[5]. All three SeeA* variants outperform the existing methods in terms of the mean ADP reduction rates against the baseline *resyn2*. SeeA*(Cluster) achieves the highest ADP reduction (i.e., 23.5%), obviously surpassing the state-of-the-art ABC-RL's 20.9%. Guided by the same heuristic, SeeA*(Cluster) outperforms the A* search in 11 out of the 12 testing circuits. As illustrated by an example of the search process in Appendix K, the nodes expanded by A* tend to concentrate on a specific branch, whereas MCTS expands across multiple branches excessively due to its enforced exploration. SeeA* achieves a good balance between A* search and MCTS, ensuring that irrelevant branches are not unduly explored.

Table 2: The ADP reduction (%) rates against the *resyn2* baseline on the MCNC testing datasets.

| Algorithm | $C1$ | $C2$ | $C3$ | $C4$ | $C5$ | $C6$ | $C7$ | $C8$ | $C9$ | $C10$ | $C11$ | $C12$ | Mean ↑ |
|---|---|---|---|---|---|---|---|---|---|---|---|---|---|
| DRiLLS [29] | 18.9 | 6.7 | 8.0 | 13.0 | 38.4 | 19.1 | 5.4 | 18.0 | 14.3 | 18.6 | 6.6 | 11.0 | 14.8 |
| Online-RL [69] | 20.6 | 6.6 | 8.1 | 13.5 | 39.4 | 21.0 | 5.0 | 17.9 | 16.2 | 20.2 | 4.7 | 11.4 | 15.4 |
| SA+Pred. [10] | 17.6 | 17.0 | 15.6 | 13.0 | 46.5 | 18.2 | 8.5 | 23.6 | 19.9 | 17.6 | 10.0 | 20.3 | 19.0 |
| MCTS [37] | 17.1 | 15.9 | 13.1 | 13.0 | 46.9 | 14.9 | 6.5 | 23.2 | 17.7 | 20.5 | 13.1 | 19.7 | 18.5 |
| ABC-RL[9] | 19.9 | 19.6 | 16.8 | 15.0 | 46.9 | 19.1 | 12.1 | 24.3 | 21.3 | 21.1 | 13.6 | 21.6 | 20.9 |
| A* search | 18.3 | 16.6 | 19.7 | 15.7 | 43.6 | 15.2 | 13.3 | **25.5** | 19.4 | 20.8 | 7.5 | 18.8 | 19.5 |
| $\varepsilon$-Greedy($\varepsilon = 0.1$)[52] | 18.3 | 16.6 | 20.9 | 15.6 | 45.8 | 19.9 | 11.5 | 24.5 | 19.4 | 20.8 | 15.3 | 18.7 | 20.6 |
| PV-MCTS | 17.3 | 20.0 | 27.9 | 20.1 | 27.3 | 20.7 | 13.5 | 24.7 | 14.3 | 14.1 | 14.7 | 20.0 | 19.5 |
| PHS [39] | 21.4 | 17.1 | 11.7 | 8.4 | 47.9 | 5.2 | 8.7 | 10.2 | 20.5 | 12.0 | 7.3 | 20.8 | 15.9 |
| SeeA*(Uniform) | 21.9 | 18.7 | 21.9 | 16.5 | 37.2 | 13.8 | 12.3 | 25.5 | 21.5 | 24.1 | 21.5 | 24.0 | 21.6 |
| SeeA*(Cluster) | 23.2 | 20.8 | 22.7 | 16.2 | 45.9 | 22.6 | 13.4 | 24.8 | 22.4 | 24.2 | 20.3 | 25.1 | 23.5 |
| SeeA*(UCT) | 20.2 | 16.6 | 25.3 | 17.8 | 46.4 | 25.5 | 10.6 | 24.4 | 18.0 | 28.7 | 17.5 | 23.6 | 22.5 |

## 5.3 Results on Sokoban and path finding

The first 50000 training problems and the 1000 test problems are collected from Boxoban [23]. They are utilized to train a cost estimator and evaluate the search algorithms, respectively. More training details are provided in Appendix L. During testing, the search process is terminated with failure if the running time exceeds 10 minutes. SeeA* has successfully solved all 1000 test Sokoban cases. Notably, the solutions generated by SeeA* exhibit not only shorter lengths compared to other search algorithms such as A* search, WA*, LevinTS, and PHS but also shorter lengths than the

---

[5]The MCTS in [37] did not utilize any guiding heuristics, and fast rollout was employed for node evaluation. PV-MCTS and PHS were guided by the policy from ABC-RL [9] and the value function trained in this paper.

state-of-the-art DeepCubeA [1] algorithm. Detailed results are summarized in Appendix M. To illustrate the effectiveness of SeeA$^*$ on problems where accurate heuristics could exist but the guiding heuristic used is unreliable, experiments on path finding are conducted. A$^*$ and SeeA$^*$ exhibit similar performance when the guidance heuristic is reliable enough. However, SeeA$^*$ demonstrates significant advantages over A$^*$ when the heuristic is unreliable. More details are available in Appendix N.

## 5.4 The impact of the hyperparameters on the performance

The effects of three hyperparameters in SeeA$^*$ are empirically investigated below, i.e., the number of candidate nodes $K$, the number of clusters $N_c$, and the adjustable weight $c_b$ in Equation 4. Experiments are conducted on the USPTO benchmark for the retrosynthesis planning problem. The number $K$ is a critical parameter controlling the extent of exploration of SeeA$^*$. When $K = 1$, the node to be expanded is solely determined by the selective sampling strategy, where the heuristic function has no impact on the selection. When $K$ is too large, all opening nodes will be finally chosen as candidates because every node has a positive chance to be selected by the sampling strategy. In this case, SeeA$^*$ degenerates back to A$^*$ which highly depends on the heuristic function. When $K$ is at an appropriate range, the sampling scheme endows SeeA$^*$ with helpful exploratory capability. It is observed from Figure 11 in Appendix O that a wide range of $K$ enables SeeA$^*$ to obtain superior performance. For the extreme cases, SeeA$^*(K = 1)$ has the lowest success rate and longest solution length, and the performance of SeeA$^*(K = \infty)$, which is equivalent to A$^*$, is also discounted.

According to the results in Figure 12 in Appendix O, the performance of the clustering sampling strategy is generally very robust against the choices of $N_c$. An inadequate number of clusters makes it towards uniform sampling by ignoring the differences among the nodes, while an excessive cluster number will distract the sampling process by noise in the node representation learning. The hyperparameter $c_b$ controls the balance between exploration and exploitation in the UCT-like sampling strategy. A large $c_b$ favors exploration during the selection of candidate nodes. From Figure 13 in Appendix O, either too large or too small $c_b$ are detrimental to the efficiency of SeeA$^*$, and the UCT-like sampling strategy achieves excellent results when $c_b$ is in the range of $[0.15, 0.4]$.

## 6 Conclusion

In this paper, the SeeA$^*$ search is proposed to enhance the exploration behavior of the A$^*$ search by selecting expanded nodes from the sampled candidate nodes, rather than the entire set of open nodes. A node that is evaluated not to have the best estimated heuristic value may be selected and explored, thereby jumping out of the local optimum induced by inaccuracies in the heuristic function. Three sampling strategies are presented in the paper. Furthermore, we have theoretically established that SeeA$^*$ is more efficient than A$^*$ search when the estimation of heuristic functions is not accurate enough. Experiments on two diverse real-world applications in chemistry and circuit design and one puzzle-solving game demonstrate the efficiency of SeeA$^*$.

If the model exhibits precise state evaluation, the incorporation of exploration into A$^*$ search becomes redundant. However, in practical applications, where problems tend to be intricate or lack sufficient training data, obtaining accurately predictive heuristic functions is challenging. As suggested in Equation 7, in addition to reducing the prediction error $\sigma$, the probability of expanding the optimal nodes is also improved by using a smaller number of candidate nodes $K$ to include the optimal node in the candidate set with a greater likelihood $P(n_1 \in \mathcal{D})$. Screening candidate nodes reduces the search space, thereby enhancing search efficiency. Investigations on more effective sampling strategies will be conducted in future work. SeeA$^*$ will contribute to solving practical problems with limited samples. However, this work is still in the nascent stages without further applications related to people's daily lives currently, and thus there are no immediate ethical or harmful social impacts.

## 7 Acknowledgement

This work was supported by the National Natural Science Foundation of China (grants No. 62172273) and the Shanghai Municipal Science and Technology Major Project, China (Grant No. 2021SHZDZX0102).

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

# A  Pseudocodes for SeeA* and its sampling strategies

In this section, pseudocode for the SeeA* search algorithm, and three sampling strategies: uniform sampling strategy, clustering sampling strategy, and UCT-like sampling strategy, are given. Algorithm 1 provides a comprehensive summary of the computational process of SeeA*. The algorithm begins by initializing the Open set $\mathcal{O}$ with the initial node $n_0$. In each step, a candidate set $\mathcal{D}$ is obtained using the sampling strategy $F$. The node with the minimum $f$ value from the candidate set is selected to be expanded. If the algorithm reaches a goal state, it successfully finds a solution and terminates. Otherwise, the expanded node is moved to the Closed set $\mathcal{C}$, and its child nodes are added to the Open set $\mathcal{O}$. This process is repeated until the $\mathcal{O}$ becomes empty.

The uniform sampling strategy is presented in Algorithm 2. If the size of the open set $|\mathcal{O}|$ does not exceed the required number of candidate nodes to be sampled, all nodes in the open set are considered as candidate nodes. However, if the size exceeds the required number, $K$ nodes are uniformly sampled from the open set to serve as candidate nodes.

---

**Algorithm 1:** SeeA* search algorithm

---

**Input:** root node $n_0$, sampling strategy $F$, the number of candidate nodes $K$, the maximum number of expansions $T_{max}$, and heuristic evaluation function $f$.

Initialize $\mathcal{C} \leftarrow \emptyset$, $\mathcal{O} \leftarrow \{n_0\}$, and $t \leftarrow 0$.

**repeat**

    Prepare candidate set $\mathcal{D} = F(\mathcal{O}, K)$.

    $n \leftarrow \arg\min_{n \in \mathcal{D}} f(n)$.

    **if** $n$ is the goal node **then**

        **return** $n$

    **else**

        $\mathcal{C} \leftarrow \mathcal{C} \cup \{n\}$, $\mathcal{O} \leftarrow \mathcal{O} \cup CH(n) \setminus \{n\}$.

    **end if**

    $t \leftarrow t + 1$

**until** $\mathcal{O}$ is empty or $t \geq T_{max}$

**return** False

---

**Algorithm 2:** Uniform sampling strategy

---

**Input:** open set $\mathcal{O}$, the number of candidate nodes $K$.

**if** $|\mathcal{O}| \leq K$ **then**

    **return** $\mathcal{O}$

**else**

    $\mathcal{D} \leftarrow$ Uniformly sample $K$ nodes from $\mathcal{O}$

    **return** $\mathcal{D}$

**end if**

---

**Algorithm 3:** Competitive clustering sampling strategy

---

**Input:** clusters $S^c = \{S_1^c, S_2^c, \cdots, S_{N_c}^c\}$, the number of candidate nodes $K$.

Initialize $\mathcal{D} \leftarrow \emptyset$

$K_e \leftarrow \lceil \frac{K}{|S^c|} \rceil$

**for** $S_i^c \in S^c$ **do**

    **if** $|S_i^c| \leq K_e$ **then**

        $\mathcal{D}_i \leftarrow S_i^c$

    **else**

        $\mathcal{D}_i \leftarrow$ Uniformly sample $K_e$ nodes from $\mathcal{S}_i^c$

    **end if**

    $\mathcal{D} \leftarrow \mathcal{D} \cup \mathcal{D}_i$

**end for**

**return** $\mathcal{D}$

---

The clustering sampling strategy is summarized in Algorithm 3. The open nodes are divided into $N_c$ clusters, and an equal number of candidate nodes are sampled from each cluster. To sample $K$

candidate nodes, each cluster requires sampling $K_c = \lceil \frac{K}{|S^c|} \rceil$ nodes. For each cluster, if the number of open nodes within this cluster does not exceed $K_c$, all nodes are selected as candidate nodes. Otherwise, $K_c$ nodes are uniformly sampled as candidate nodes. To reduce computational overhead, competitive learning is employed for clustering, enabling avoidance of the need for re-clustering when new open nodes are added after each expansion. The algorithm for this competitive clustering process is displayed in Algorithm 5. $N_c$ clustering centers are initialized randomly. When a new node is added to the open set, it is assigned to the cluster whose center is closest to it, and the coordinates of that cluster center are updated by moving toward the position of the newly added node.

UCT-like sampling strategy is provided in Algorithm 4, respectively. Each node $n$ is evaluated by a statistic $E(n)$ in Equation 4, which can achieve a balance between exploitation and exploration. The $f$ value of the node is used to evaluate the quality and the depth $d(n)$ is used to encourage exploration. If the size of the open set $|\mathcal{O}|$ does not exceed the required number of candidate nodes to be sampled, all nodes in the open set are considered as candidate nodes. Otherwise, the $K$ nodes with the smallest $E$ values are selected as the candidate nodes.

---

**Algorithm 4:** UCT-like sampling strategy

---

**Input:** open set $\mathcal{O}$, the number of candidate nodes $K$, hyperparameter $c_b$.
**if** $|\mathcal{O}| \leq K$ **then**
  **return** $\mathcal{O}$
**else**
  $d_{max} \leftarrow \max_{n \in \mathcal{O}} d(n)$
  **for** $n \in \mathcal{O}$ **do**
    Calculate $E(n)$ with Equation 4.
  **end for**
  $\mathcal{D} \leftarrow \{n | E(n) \text{ is the } K \text{ smallest values in } \mathcal{O}\}$
  **return** $\mathcal{D}$
**end if**

---

**Algorithm 5:** Node expanding for competitive clustering

---

**Input:** clusters $S^c = \{S_1^c, S_2^c, \cdots, S_{N_c}^c\}$ and their centers $W^c = \{w_1^c, w_2^c, \cdots, w_{N_c}^c\}$, expanded node $n$, feature extraction function $f_h$, and weight $\eta$.
**for** $n \in \mathcal{CH}(n)$ **do**
  $w \leftarrow f_h(n)$
  $j \leftarrow \arg\min_{w_i^c \in W^c} ||w - w_i^c||$
  $S_j^c \leftarrow S_j^c \cup \{n\}$
  $w_j^c \leftarrow w_j^c + \eta \times (w - w_j^c)$
**end for**
**return** $S^c, W^c$

---

## B  An example of the clustering sampling strategy

As shown in Figure 2 (a), the sampled nodes exhibit a uniform distribution overall, resulting in a proportional relationship between the number of sampled nodes within each cluster and the total number of nodes within each class. Clusters with fewer nodes are highly susceptible to the occurrence of no nodes being sampled as candidate nodes under limited sampling size. Therefore, this sampling strategy cannot effectively explore the various potential expansion directions. As depicted in Figure 2 (b), at least one node from each cluster is sampled compulsorily. Consequently, the probability of including nodes on the optimal branch is increased, thereby facilitating search efficiency.

## C  Monotonicity of $p_\sigma$

$p_\sigma$ is the probability of $P(f(n) < f(n'))$, in which $n$ is the node on the optimal path and $n'$ is the node on the non-optimal path. According to Assumption 4.1:

$$P(f(n)|\sigma) \sim \mathcal{U}(\mu_0^f - \sigma, \mu_0^f + \sigma) \tag{12}$$

$$P(f(n')|\mu_s, \sigma) \sim \mathcal{U}(f^*(n') - \sigma, f^*(n') + \sigma) \tag{13}$$

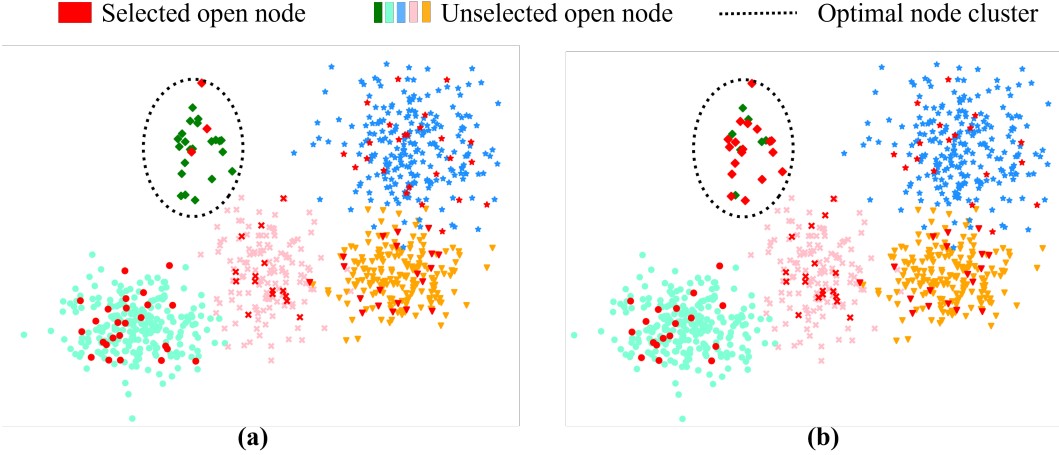

Figure 2: Set of selected candidate nodes $\mathcal{D}$ obtained by (a) uniform sampling strategy; (b) clustering sampling strategy.

$$f^*(n') \sim \mathcal{G}(\mu_1^f, \sigma_s^2), \mu_1^f > \mu_0^f \tag{14}$$

If $\mu_0^f + \sigma \leq f^*(n') - \sigma$:

$$P\left(f(n) \leq f(n')|f^*(n') \geq \mu_0^f + 2\sigma, \sigma\right) = 1 \tag{15}$$

If $\mu_0^f - \sigma \geq f^*(n') + \sigma$:

$$P\left(f(n) \leq f(n')|f^*(n') \leq \mu_0^f - 2\sigma, \sigma\right) = 0 \tag{16}$$

If $\mu_0 - 2\sigma \leq f^*(n') < \mu_0$:

$$P\left(f(n) \leq f(n')|f^*(n'), \sigma\right) = \int_{\mu_0^f - \sigma}^{f^*(n') + \sigma} \frac{1}{2\sigma} \int_{f(n)}^{f^*(n') + \sigma} \frac{1}{2\sigma} df(n') df(n) \tag{17}$$

$$= \frac{1}{4\sigma^2} \int_{\mu_0^f - \sigma}^{f^*(n') + \sigma} (f^*(n') + \sigma - f(n)) df(n) = \frac{\left(f^*(n') - \mu_0^f + 2\sigma\right)^2}{8\sigma^2}$$

If $\mu_0^f \leq f^*(n') < \mu_0^f + 2\sigma$:

$$P\left(f(n) \leq f(n')|f^*(n'), \sigma\right) = \int_{\mu_0^f - \sigma}^{f^*(n') - \sigma} \frac{1}{2\sigma} \int_{f^*(n') - \sigma}^{f^*(n') + \sigma} \frac{1}{2\sigma} df(n') df(n)$$

$$+ \int_{f^*(n') - \sigma}^{\mu_0^f + \sigma} \frac{1}{2\sigma} \int_{f(n)}^{f^*(n') + \sigma} \frac{1}{2\sigma} df(n') df(n) \tag{18}$$

$$\int_{\mu_0^f - \sigma}^{f^*(n') - \sigma} \frac{1}{2\sigma} \int_{f^*(n') - \sigma}^{f^*(n') + \sigma} \frac{1}{2\sigma} df(n') df(n)$$

$$= \frac{1}{4\sigma^2}(f^*(n') + \sigma - f^*(n') + \sigma)(f^*(n') - \sigma - \mu_0^f + \sigma)$$

$$= \frac{2\sigma(f^*(n') - \mu_0^f)}{4\sigma^2} = \frac{f^*(n') - \mu_0^f}{2\sigma} \tag{19}$$

$$\int_{f^*(n')-\sigma}^{\mu_0^f+\sigma} \frac{1}{2\sigma} \int_{f(n)}^{f^*(n')+\sigma} \frac{1}{2\sigma} df(n') df(n)$$

$$= \int_{f^*(n')-\sigma}^{\mu_0^f+\sigma} \frac{f^*(n') + \sigma - f(n)}{4\sigma^2} df(n) = \frac{1}{2} - \frac{\left(f^*(n') - \mu_0^f\right)^2}{8\sigma^2} \tag{20}$$

$$P\left(f(n) \le f(n')|f^*(n'), \sigma\right) = \frac{f^*(n') - \mu_0^f}{2\sigma} + \frac{1}{2} - \frac{\left(f^*(n') - \mu_0^f\right)^2}{8\sigma^2}$$

$$= 1 - \frac{\left(f^*(n') - \mu_0^f - 2\sigma\right)^2}{8\sigma^2} \tag{21}$$

In summary:

$$P\left(f(n) \le f(n')|f^*(n'), \sigma\right) = \begin{cases} 1, & f^*(n') \ge \mu_0^f + 2\sigma \\ 1 - \dfrac{\left(f^*(n') - \mu_0^f - 2\sigma\right)^2}{8\sigma^2}, & \mu_0^f \le f^*(n') < \mu_0^f + 2\sigma \\ \dfrac{\left(f^*(n') - \mu_0^f + 2\sigma\right)^2}{8\sigma^2}, & \mu_0^f - 2\sigma \le f^*(n') < \mu_0^f \\ 0, & f^*(n') \le \mu_0^f - 2\sigma \end{cases} \tag{22}$$

We have that:

$$P\left(f(n) \le f(n')|\sigma\right) = \int P\left(f(n) \le f(n')|f^*(n'), \sigma\right) P(f^*(n')) df^*(n'), \tag{23}$$

$$f^*(n') \sim \mathcal{G}(\mu_1^f, \sigma_s^2) \tag{24}$$

then we want to prove that $P\left(f(n) \le f(n')\right)$ decreases with $\sigma$. Let $m(f^*(n')|\sigma) = P\left(f(n) \le f(n')|f^*(n'), \sigma\right)$, we first prove that $m(f^*(n')|\sigma)$ centrally symmetric about $(\mu_0^f, \frac{1}{2})$ by proving that $m\left(f^*(n')|\sigma\right) = 1 - m\left(2\mu_0^f - f^*(n')|\sigma\right)$:

If $f^*(n') \ge \mu_0^f + 2\sigma$:

$$m(f^*(n')|\sigma) = 1 \tag{25}$$

$$2\mu_0^f - f^*(n') \le \mu_0^f - 2\sigma \tag{26}$$

$$m\left(2\mu_0^f - f^*(n')|\sigma\right) = 0 = 1 - m\left(f^*(n')|\sigma\right) \tag{27}$$

If $\mu_0^f \le f^*(n') < \mu_0^f + 2\sigma$:

$$m(\mu_s^f|\sigma) = 1 - \frac{\left(f^*(n') - \mu_0^f - 2\sigma\right)^2}{8\sigma^2} \tag{28}$$

$$\mu_0^f - 2\sigma < 2\mu_0^f - f^*(n') \le \mu_0^f \tag{29}$$

$$m\left(2\mu_0^f - f^*(n')|\sigma\right) = \frac{\left(2\mu_0^f - f^*(n') - \mu_0^f + 2\sigma\right)^2}{8\sigma^2}$$

$$= \frac{\left(f^*(n') - \mu_0^f - 2\sigma\right)^2}{8\sigma^2} = 1 - m(f^*(n')|\sigma) \tag{30}$$

If $\mu_0^f - 2\sigma \leq f^*(n') < \mu_0^f$:

$$m(f^*(n')|\sigma) = \frac{\left(f^*(n') - \mu_0^f + 2\sigma\right)^2}{8\sigma^2} \tag{31}$$

$$\mu_0^f < 2\mu_0^f - f^*(n') \leq \mu_0^f + 2\sigma \tag{32}$$

$$m\left(2\mu_0^f - f^*(n')|\sigma\right) = 1 - \frac{\left(2\mu_0^f - f^*(n') - \mu_0^f - 2\sigma\right)^2}{8\sigma^2}$$

$$= 1 - \frac{\left(f^*(n') - \mu_0^f + 2\sigma\right)^2}{8\sigma^2} = 1 - m(f^*(n')|\sigma) \tag{33}$$

If $f^*(n') \leq \mu_0^f - 2\sigma$:

$$m(f^*(n')|\sigma) = 0 \tag{34}$$

$$2\mu_0^f - f^*(n') \geq \mu_0^f + 2\sigma \tag{35}$$

$$m\left(2\mu_0^f - f^*(n')|\sigma\right) = 1 = 1 - m(f^*(n')|\sigma) \tag{36}$$

Therefore, $m(f^*(n')|\sigma)$ centrally symmetric about $(\mu_0^f, \frac{1}{2})$. Then we want to illustrate the monotonicity of the function $m(\mu_s^f|\sigma)$ about the variable $\sigma$, and $\sigma > 0$

If $f^*(n') \geq \mu_0^f + 2\sigma$ or $f^*(n') \leq \mu_0^f - 2\sigma$, $m(f^*(n')|\sigma)$ is constant. If $\mu_0^f - 2\sigma \leq f^*(n') < \mu_0^f$, $m(f^*(n')|\sigma)$ is monotonic increasing, because

$$\mu_0^f - 2\sigma \leq f^*(n') < \mu_0^f \Rightarrow \sigma \geq \frac{\mu_0^f - \mu_s^f}{2} \tag{37}$$

$$\left[\frac{\left(f^*(n') - \mu_0^f + 2\sigma\right)^2}{8\sigma^2}\right]' = \frac{\left(\mu_0^f - f^*(n')\right)\left(2\sigma - \mu_0^f + f^*(n')\right)}{4\sigma^3} \geq 0 \tag{38}$$

If $\mu_0^f \leq f^*(n') < \mu_0^f + 2\sigma$, $m(f^*(n')|\sigma)$ is monotonic decreasing, because

$$\mu_0^f \leq f^*(n') < \mu_0^f + 2\sigma \Rightarrow \sigma > \frac{f^*(n') - \mu_0^f}{2}$$

$$\left[1 - \frac{\left(f^*(n') - \mu_0^f - 2\sigma\right)^2}{8\sigma^2}\right]' = \frac{\left(f^*(n') - \mu_0^f\right)\left(f^*(n') - \mu_0^f - 2\sigma\right)}{4\sigma^3} < 0 \tag{39}$$

Therefore, as shown in Figure 3, $P\left(f(n) \leq f(n')|f^*(n'), \sigma\right)$ is not always decrease with $\sigma$. However, on average, $P\left(f(n) \leq f(n')|\sigma\right)$ will always decrease with $\sigma$, and we will prove this.

For convenience, let $F(\sigma) = P(f(n) \leq f(n')|\sigma)$. Assume $\sigma_1 < \sigma_2$, then:

$$F(\sigma_1) - F(\sigma_2) = \int_{-\infty}^{\infty} \left[m(f^*(n')|\sigma_1) - m(f^*(n')|\sigma_2)\right] P(f^*(n'))df^*(n')$$

$$= \int_{-\infty}^{\mu_0^f} \left[m(f^*(n')|\sigma_1) - m(f^*(n')|\sigma_2)\right] P(f^*(n'))df^*(n')$$

$$+ \int_{\mu_0^f}^{\infty} \left[m(f^*(n')|\sigma_1) - m(f^*(n')|\sigma_2)\right] P(f^*(n'))df^*(n') \tag{40}$$

Because the function $m$ is centrally symmetric about $\left(\mu_0^f, \frac{1}{2}\right)$:

$$m(f^*(n')|\sigma) = 1 - m(2\mu_0^f - f^*(n')|\sigma) \tag{41}$$

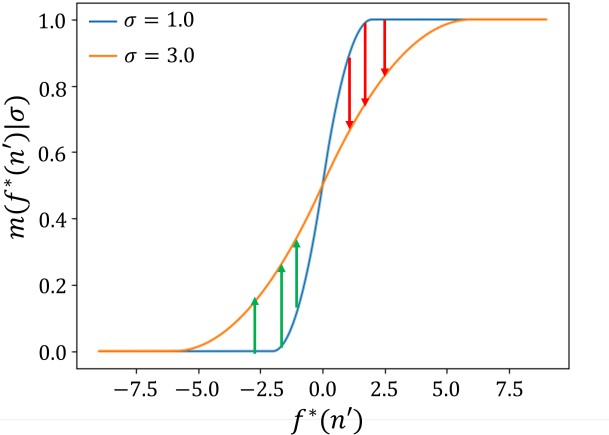

Figure 3: Example for the monotonicity of $P\left(f(n) \leq f(n')|f^*(n'), \sigma\right)$.

Therefore:

$$\int_{-\infty}^{\mu_0^f} \left[m(f^*(n')|\sigma_1) - m(f^*(n')|\sigma_2)\right] P(f^*(n'))df^*(n')$$

$$= \int_{-\infty}^{\mu_0^f} \left[m(2\mu_0^f - f^*(n')|\sigma_2) - m(2\mu_0^f - f^*(n')|\sigma_1)\right] P(f^*(n'))df^*(n')$$

$$= \int_{\mu_0^f}^{\infty} \left[m(f^*(n')|\sigma_2) - m(f^*(n')|\sigma_1)\right] P(2\mu_0^f - f^*(n'))df^*(n') \tag{42}$$

$$F(\sigma_1) - F(\sigma_2)$$
$$= \int_{\mu_0^f}^{\infty} \left[m(f^*(n')|\sigma_1) - m(f^*(n')|\sigma_2)\right] \left(P(f^*(n')) - P(2\mu_0^f - f^*(n'))\right) df^*(n') \tag{43}$$

When $f^*(n') > \mu_0^f$, $m(f^*(n')|\sigma)$ is monotonic decreasing with $\sigma$:

$$m(f^*(n')|\sigma_1) - m(f^*(n')|\sigma_2) > 0 \tag{44}$$

Because $f^*(n') \sim \mathcal{G}(\mu_1^f, \sigma_s^2)$, if $P(f^*(n')) < P(2\mu_0^f - f^*(n'))$,

$$|f^*(n') - \mu_1^f| > |2\mu_0^f - f^*(n') - \mu_1^f| \tag{45}$$

$$\left(f^*(n') - \mu_1^f\right)^2 > \left(2\mu_0^f - f^*(n') - \mu_1^f\right)^2 \tag{46}$$

$$4\left(\mu_0^f - \mu_1^f\right)\left(f^*(n') - \mu_0^f\right) > 0 \tag{47}$$

Assumption 4.1 gives $\mu_1^f > \mu_0^f$, and Equation 43 only considers the situation when $f^*(n') > \mu_0^f$. Therefore Equation 47 is not established and $P(f^*(n')) \geq P(2\mu_0^f - f^*(n'))$. Therefore:

$$P(f^*(n')) - P(2\mu_0^f - f^*(n')) \geq 0 \tag{48}$$

$$F(\sigma_1) - F(\sigma_2) > 0, \quad \sigma_1 < \sigma_2 \tag{49}$$

Therefore, $F(\sigma)$ is monotonic decreasing. The larger the estimation error $\sigma$, the less the probability of $P\left(f(n) < f(n')|\sigma\right)$.

The assumption in Corollary 4.2 that the prediction error for $f^*$ is uniformly distributed is quite strong. To further illustrate the applicability of the algorithm, we also prove that Corollary 4.2 is established if the noise follows a Gaussian distribution. Denoting Gaussian distribution as $\mathcal{G}(\cdot, \cdot)$, the assumption regarding the distribution of the estimated values will be adjusted to:

**Assumption C.1** *For each node $n$ on the optimal path, $f(n) \sim \mathcal{G}(\mu_0^f, \sigma^2)$. For nodes not on the optimal path, $f(n) \sim \mathcal{G}(f^*(n), \sigma^2)$, and $f^*(n)$ are independently and identically sampled from $\mathcal{G}(\mu_1^f, \sigma_s^2)$. $\mu_0^f < \mu_1^f$ holds because the optimal path has a lower cost.*

For two Gaussian distributions, we have the following lemma [62, 67]:

**Lemma C.2** *Assume $x \sim \mathcal{G}(\mu_1, \sigma_1^2)$, $y \sim \mathcal{G}(\mu_2, \sigma_2^2)$. If $x$, $y$ are independent of each other and $\mu_2 > \mu_1$, then*

$$P(x > y) = \frac{1}{\pi} \int_0^{\frac{\pi}{2}} \exp \left\{ -\frac{1}{2} \frac{[(\mu_2 - \mu_1)/\sqrt{\sigma_1^2 + \sigma_2^2}]^2}{\cos^2 \theta} \right\} d\theta. \tag{50}$$

For a node $n$ on the optimal path, $f(n) \sim \mathcal{G}(\mu_0^f, \sigma^2)$. For a node $n'$ off the optimal path, $f(n') \sim \mathcal{G}(f^*(n'), \sigma^2)$. If $\mu_0^f > f^*(n')$:

$$P(f(n) < f(n') | \mu_0^f > f^*(n')) = \frac{1}{\pi} \int_0^{\frac{\pi}{2}} \exp \left\{ -\frac{1}{2} \frac{(f^*(n') - \mu_0^f)^2}{2\sigma^2 \cos^2 \theta} \right\} d\theta = m(f^*(n')|\sigma) \tag{51}$$

Otherwise:

$$P(f(n) < f(n') | \mu_0^f < f^*(n')) = 1 - \frac{1}{\pi} \int_0^{\frac{\pi}{2}} \exp \left\{ -\frac{1}{2} \frac{(f^*(n') - \mu_0^f)^2}{2\sigma^2 \cos^2 \theta} \right\} d\theta$$
$$= 1 - m(f^*(n')|\sigma). \tag{52}$$

The probability that the $f$ value of the optimal node is less than the $f$ value of a non-optimal node is

$$F(\sigma) = P(f(n) < f(n')|\sigma) = \int_{f^*(n') < \mu_0^f} P(f^*(n')) m(f^*(n')|\sigma) df^*(n')$$
$$+ \int_{f^*(n') \geq \mu_0^f} P(f^*(n'))(1 - m(f^*(n')|\sigma)) df^*(n'). \tag{53}$$

If $\sigma_2 > \sigma_1$:

$$F(\sigma_2) - F(\sigma_1) = \int_{f^*(n') < \mu_0^f} P(f^*(n'))(m(f^*(n')|\sigma_2) - m(f^*(n')|\sigma_1)) df^*(n')$$
$$+ \int_{f^*(n') \geq \mu_0^f} P(f^*(n'))(m(f^*(n')|\sigma_1) - m(f^*(n')|\sigma_2)) df^*(n'). \tag{54}$$

$m(f^*(n')|\sigma)$ is symmetric about the axis $f^*(n') = \mu_0^f$,

$$m(f^*(n')|\sigma) = m(2\mu_0^f - f^*(n')|\sigma). \tag{55}$$

Equation 54 is equivalent to

$$F(\sigma_2) - F(\sigma_1) = \int_{f^*(n') \geq \mu_0^f} (P(2\mu_0^f - f^*(n')) - P(f^*(n')))$$
$$\times (m(f^*(n')|\sigma_2) - m(f^*(n')|\sigma_1)) df^*(n') \tag{56}$$

According to the definition, $m$ is monotonically increasing with respect to $\sigma$. Therefore, $m(f^*(n')|\sigma_2) - m(f^*(n')|\sigma_1) > 0$. Because $f^*(n') \sim \mathcal{N}(\mu_1^f, \sigma_2^2)$ and $\mu_0^f < \mu_1^f$, we have $P(2\mu_0^f - f^*(n')) - P(f^*(n')) < 0$ when $f^*(n') \geq \mu_0^f$. Therefore, $F(\sigma_2) - F(\sigma_1) < 0$ is established, and $P(f(n) < f(n')|\sigma)$ decreases as the prediction error $\sigma$ increases when the noise is Gaussian distribution. The above analyses will be added to the revised paper to further elucidate the impact of prediction errors. Under both the uniform error distribution and the Gaussian error distribution, the larger the prediction error, the lower the likelihood of selecting the optimal node.

# D  Network architecture of policy and value in retrosynthesis planning

Each molecule is encoded using a 2048-dimensional Morgan Fingerprint vector [45] as the input for the heuristic functions. The policy network is a multi-class task based on chemical reaction templates. 381302 templates are available. The policy network architecture is summarized as follows:

- A fully connected layer with dimensions $[2048, 512]$.

- A batch normalization layer.

- A dropout layer with a dropout rate of $0.3$.

- Another fully connected layer with dimensions $[512, 381302]$.

- A softmax layer.

The top 50 reaction templates with the highest probabilities are retained, and corresponding chemical reactions are generated using rdchiral package [12], an RDKit [32] wrapper for handling stereochemistry in retrosynthetic template extraction and application.

The output of the value network is a scalar to estimate the synthetic cost of the input molecules. The architecture is summarized as follows:

- A fully connected layer with dimensions $[2048, 128]$.

- A ReLU activation layer.

- A dropout layer with a dropout rate of $0.1$.

- A fully connected layer with dimensions $[128, 1]$.

- Normalize the output $y$ with $log(1 + e^y)$.

Parameters of the policy and value network are the same with Retro$^*$+ [30]. No fine-tuning has been performed on the network.

# E  Introduction of test molecule datasets

Besides the USPTO benchmark, six additional datasets comprising 4719 molecules are employed for robust validation. The introduction of these datasets is as follows.

**logP** [7]: The logarithm of the partition coefficient (logP) is a measure of the solubility of a molecule in a particular solvent. The logS of a molecule can affect the molecule's pharmacokinetics and pharmacodynamics.

**logS** [60]: It is used to evaluate the solubility of molecules, which can affect the absorption, distribution, metabolism, and excretion (ADME) of a drug candidate.

**Toxicity LD50** [56]: Toxicity plays a key role in determining the safety and efficacy of drugs.

**Ames** [25]: The Ames test is commonly used in the field of drug development to evaluate the potential mutagenicity of drug candidates, as well as other chemicals that may be used in drug manufacturing or as excipients.

**BBBP** [35]: Blood-brain barrier (BBB) is a protective barrier that separates the bloodstream from the brain to prevent harmful substances from entering the brain. BBB penetration (BBBP) is considered when developing new drugs.

**ClinTox** [21]: It is a dataset collecting drugs approved by the FDA and drugs that have failed clinical trials for toxicity reasons.

To clean these datasets, molecules present in either the USPTO database or the building block set are removed. Additionally, molecules that can be solved by Retro$^*$ in one step and those that can be easily solved by a heuristic-based BFS planning algorithm within a fixed time limit are also excluded. After processing, 4719 molecules are retained.

# F    Search tree representation in retrosynthesis planning

The search tree employed in this paper in retrosynthesis planning problem is displayed in Figure 4. The initial state contains only the target molecule. The edges in the search tree represent the chemical reactions that enable state transitions between the connected nodes, which decompose the input product into its reactants. A state is a set of molecules that are able to synthesize the target molecule, which is decomposed from the target molecule along the traverse reaction path from the root to this node. The retrosynthetic planning problem is solved if all molecules within a state are available building blocks. For A single-step retrosynthetic prediction model is utilized as the policy model to generate potential chemical reactions yielding the input molecule. For non-terminal intermediate nodes, all molecules within the node are sorted in alphabetical order based on their SMILES representation. The first non-building block molecule in the sorted list is selected as the input molecule of the single-step prediction model. The number of available reactions provided by the policy for a given node corresponds to the number of branching factors within the search tree.

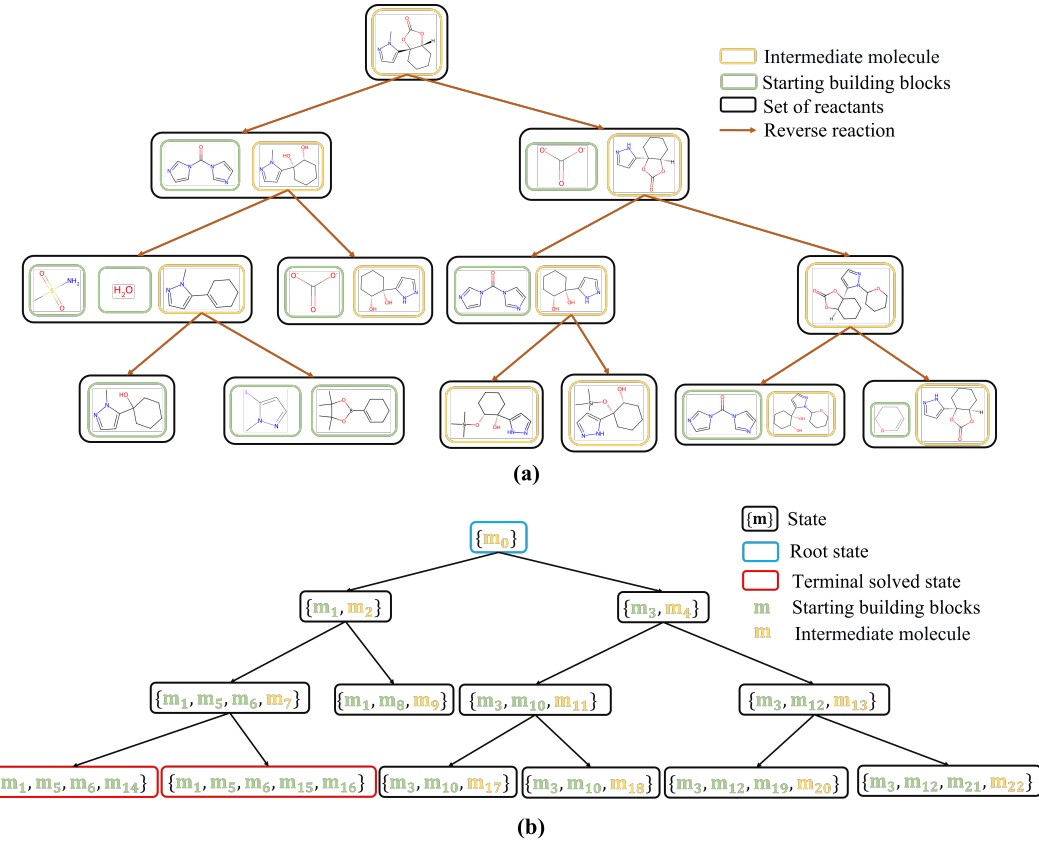

Figure 4: The process involves transforming the representation of the chemical retrosynthetic route into the search tree representation used in this paper. (a) is the real chemical retrosynthetic route, in which the reverse reaction decomposes the input product molecule into several reactant molecules; (b) is the corresponding search tree representation, and each node in the tree contains all molecules decomposed from the target molecule along the traverse reaction path from the root to this node.

Retro$^*$ and Retro$^*$+ utilize the AND-OR tree as the representation for the planning process. In this representation, a reaction is denoted by an AND node, with its child nodes representing the reactant molecules involved in that reaction. Similarly, a molecule is represented by an OR node, with its child nodes representing the chemical reactions capable of synthesizing that molecule. A chemical reaction can be taken if all of its child reactants can be synthesized, and a chemical molecule can be synthesized if there exists at least one child chemical reaction that can take place. We also applied

SeeA$^*$ to the And-OR Tree framework, and under the same settings, utilizing the uniform sampling strategy to obtain candidate nodes, we observed an increase in success rate from Retro$^*$+'s 91.05% to 92.11% in USPTO benchmark. The averafe siolution length is decreased from Retro$^*$+'s 8.74 to 8.39. This finding demonstrates the effectiveness of the algorithm in the AND-OR tree structure.

## G   Test results on each dataset for retrosynthetic planning

Average test results on the seven molecule datasets and results for the six additional testing molecule datasets are presented in Table 3 and 4 respectively. SeeA$^*$ achieves the maximum number of problem-solving instances and obtains the shortest solution length across all datasets.

Table 3: Test success accuracy on the seven dataset for retrosynthetic planning problem (%).

| Algorithm | USPTO | logP | logS | Toxicity LD50 | Ames | BBBP | ClinTox | Mean |
|---|---|---|---|---|---|---|---|---|
| Retro$^*$ | 86.84 | 53.96 | 67.08 | 55.39 | 57.40 | 47.87 | 38.69 | 54.66 |
| Retro$^*$+ | 91.05 | 61.14 | 69.29 | 59.98 | 63.51 | 52.46 | 43.15 | 59.93 |
| A$^*$ | 88.42 | 58.71 | 68.55 | 59.17 | 62.98 | 51.80 | 42.04 | 58.73 |
| WA$^*$ | 84.21 | 58.43 | 68.30 | 59.52 | 62.89 | 52.30 | 44.59 | 58.87 |
| MCTS | 89.47 | 58.15 | 67.08 | 58.26 | 63.42 | 52.95 | 46.34 | 59.20 |
| LevinTS | 96.84 | 61.14 | 70.76 | 60.32 | 64.84 | 54.92 | 43.63 | 61.01 |
| PHS | 87.37 | 55.45 | 65.60 | 57.00 | 59.96 | 50.98 | 39.01 | 56.16 |
| $\varepsilon$-Greedy | 92.11 | 61.14 | 70.02 | 62.04 | 65.01 | 54.26 | 45.22 | 61.23 |
| SeeA$^*$(Uniform) | 96.84 | 63.37 | 71.00 | 62.73 | **67.32** | 56.39 | 45.70 | 62.97 |
| SeeA$^*$(Cluster) | 98.42 | **64.12** | 72.73 | 63.53 | 66.08 | **57.54** | **47.77** | **63.56** |
| SeeA$^*$(UCT) | **98.95** | 63.93 | **72.97** | **63.65** | 65.28 | 56.89 | 47.45 | 63.31 |

Table 4: Test solution length on the seven dataset for retrosynthetic planning problem.

| Algorithm | USPTO | logP | logS | ToxicityLD50 | Ames | BBBP | ClinTox | Mean |
|---|---|---|---|---|---|---|---|---|
| Retro$^*$ | 9.71 | 16.67 | 12.63 | 16.24 | 15.91 | 18.29 | 21.11 | 16.58 |
| Retro$^*$+ | 8.74 | 15.01 | 12.26 | 15.23 | 14.67 | 17.37 | 20.06 | 15.44 |
| A$^*$ | 9.27 | 15.64 | 12.44 | 15.49 | 14.94 | 17.56 | 20.26 | 15.78 |
| WA$^*$ | 10.16 | 15.62 | 12.46 | 15.39 | 14.90 | 17.36 | 19.43 | 15.66 |
| MCTS | 8.23 | 16.27 | 13.00 | 15.99 | 15.05 | 17.35 | 19.15 | 15.91 |
| LevinTS | 7.45 | 15.55 | 12.48 | 15.74 | 15.02 | 17.25 | 20.24 | 15.74 |
| PHS | 10.19 | 16.56 | 13.29 | 16.11 | 15.72 | 17.79 | 21.09 | 16.51 |
| $\varepsilon$-Greedy | 43.78 | 23.21 | 12.76 | 16.70 | 16.32 | 18.43 | 23.82 | 19.88 |
| SeeA$^*$(Uniform) | 7.34 | 14.64 | 11.81 | 14.76 | 14.00 | 16.62 | 19.41 | 14.85 |
| SeeA$^*$(Cluster) | 6.48 | 14.05 | 11.20 | 14.21 | **13.79** | **15.85** | 18.65 | **14.31** |
| SeeA$^*$(UCT) | **6.36** | **14.05** | **11.18** | **14.11** | 13.95 | 16.01 | **18.61** | 14.33 |

## H   Introduction for logic synthesis problem

Logic synthesis is the process of transforming a hardware design at the register transfer level (RTL) into a Boolean gate-level network, which is represented by an and-inverter-graph (AIG), i.e., a netlist exclusively containing AND and NOT gates. Subsequently, a sequence of functionality-preserving transformations is applied to generate an optimized AIG. Seven operations are allowed following the work of ABC [4] and other reinforcement learning algorithms [9, 29], including balance, re-substitution, re-substitution -z, rewrite, rewrite -z, refactor, and refactor -z. The number of transformations is limited to 10, which is the same with ABC-RL [9]. Finally, post-technology mapping is performed using a 7nm technology library to obtain the final netlist, which is also generated using the ABC package. Time delay and area are estimated by ABC to evaluate the solution. An example framework of solving the logic synthesis problem is presented in Figure 5.

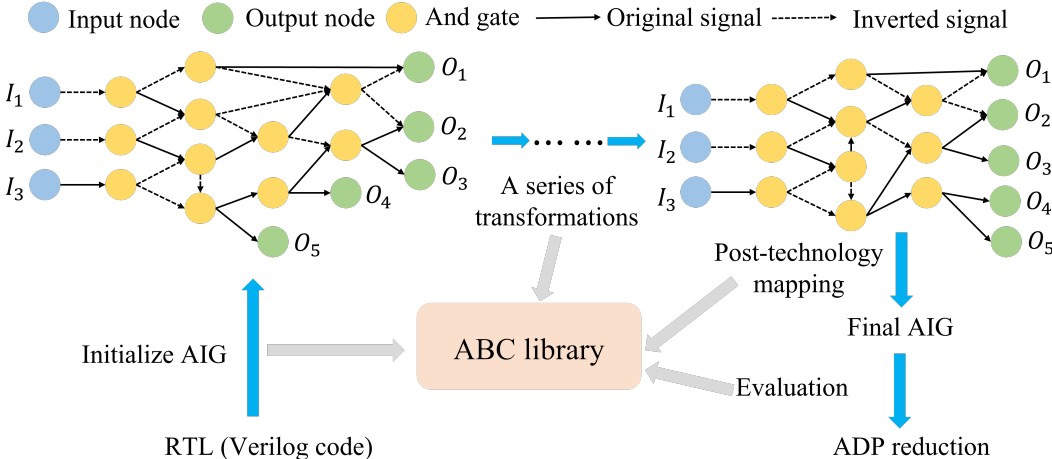

Figure 5: An example of the logic synthesis problem. The design of the hardware is represented by an and-inverter graph. After a series of transformations, a more refined AIG is obtained, while maintaining the same function as the original AIG. While preserving the input and output relationship, the number and connectivity of intermediate nodes are optimized. Post-technology mapping and final evaluation are conducted with the ABC package.

# I   Introduction of the MCNC dataset

According to the partitioning in ABC-RL [9], A total of 15 circuits from the MCNC dataset [63] are utilized as training circuits, while the remaining 12 circuits from the same dataset are employed to evaluate the performance. The number of nodes in the initial AIG of each circuit ranges from hundreds to thousands. The information about the circuits is summarized in Table 5 & 6. The number of input and output nodes varies from a few to several hundred, while the total number of nodes ranges from several hundred to thousands.

Table 5: Characterization of training circuits from MCNC dataset.

| Circuit | # Inputs | # Outputs | # Nodes | Level |
|---------|----------|-----------|---------|-------|
| alu2    | 10       | 6         | 401     | 40    |
| apex3   | 54       | 50        | 2374    | 21    |
| apex5   | 117      | 88        | 1280    | 21    |
| b2      | 16       | 17        | 1814    | 22    |
| c1355   | 41       | 32        | 504     | 26    |
| c2670   | 233      | 140       | 717     | 21    |
| c5315   | 178      | 123       | 1773    | 38    |
| c6288   | 32       | 32        | 2337    | 120   |
| frg1    | 28       | 3         | 126     | 19    |
| i7      | 199      | 67        | 904     | 6     |
| i8      | 133      | 81        | 3310    | 21    |
| m3      | 8        | 16        | 434     | 14    |
| max512  | 9        | 6         | 743     | 19    |
| prom2   | 9        | 21        | 3513    | 22    |
| table5  | 17       | 15        | 1987    | 26    |

# J   Value estimator for logic synthesis problem

The input of the value estimator consists of the initial AIG and the sequence of actions that have been taken. Nodes in the AIG (And-Inverter Graph) are represented by a two-dimensional vector, which

Table 6: Characterization of testing circuits from MCNC dataset.

| Circuit | # Inputs | # Outputs | # Nodes | Level |
|---------|----------|-----------|---------|-------|
| alu4 | 10 | 6 | 735 | 42 |
| apex1 | 45 | 45 | 2655 | 27 |
| apex2 | 39 | 3 | 445 | 29 |
| apex4 | 9 | 19 | 3452 | 21 |
| b9 | 41 | 21 | 105 | 10 |
| c880 | 60 | 26 | 327 | 24 |
| c7552 | 207 | 108 | 2074 | 29 |
| i9 | 88 | 63 | 889 | 14 |
| m4 | 8 | 16 | 760 | 14 |
| pair | 173 | 137 | 1500 | 24 |
| max1024 | 10 | 6 | 1021 | 20 |
| prom1 | 9 | 40 | 7803 | 24 |

records the node type and the number of inverted predecessors. The adjacency matrix is used to capture the node connectivity. Graph convolutional network (GCN) is used to extract the embedding for AIG, and the architecture is summarized as follows:

- A GCN (Graph Convolutional Network) layer with a hidden size of 32.
- A batch normalization layer.
- A LeakyReLU activation layer.
- A GCN layer with a hidden size of 32.
- A batch normalization layer.
- Mean pooling and max pooling are independently applied, and the outputs are concatenated to form the final embedding, which is a 64 dimension vector.

The sequence of actions and the current number of steps are combined into a string, which serves as the input to the BERT model [15] to obtain a sequence embedding with 768 dimensions. AIG embedding and sequence embedding are concatenated together as the input of the value estimator, and the architecture is:

- A fully connected layer with dimensions $[832, 256]$.
- A LeakyReLU activation layer.
- A fully connected layer with dimensions $[256, 256]$.
- A LeakyReLU activation layer.
- A fully connected layer with dimensions $[256, 1]$.
- A Tanh activation layer.

MCTS simulations are performed on the training circuits to collect samples. During the evaluation of the leaf node before backpropagation, a complete search path is obtained through a fast rollout, which is then stored as a training dataset for the value estimator. 1500 action sequences are collected for each circuit. *resyn2* synthesis recipe is used as the baseline during the evaluation. The area-delay product reduction for an action sequence $P$ is defined as

$$ADPR(AIG, P) = 1 - \frac{ADP(AIG, P)}{ADP(AIG, resyn2)}, \tag{57}$$

where ADP is $Area \times Delay$. The reward in logic synthesis problem is

$$R(AIG, P, t) = \begin{cases} \max\{-1, ADPR(AIG, P)\}, & \text{if } t = |P| \\ 0, & \text{Otherwise} \end{cases} \tag{58}$$

where $t$ is the index of the current step. The immediate reward is always $0$ except for the last step. Therefore, the ground truth value for states in the sequence $P$ are all equal to

$\max\{-1, ADPR(AIG, P)\}$. Mean square error loss is used as the learning target. Adam optimizer is employed to update the parameter with a 0.0001 learning rate. The learning process is presented in Figure 6.

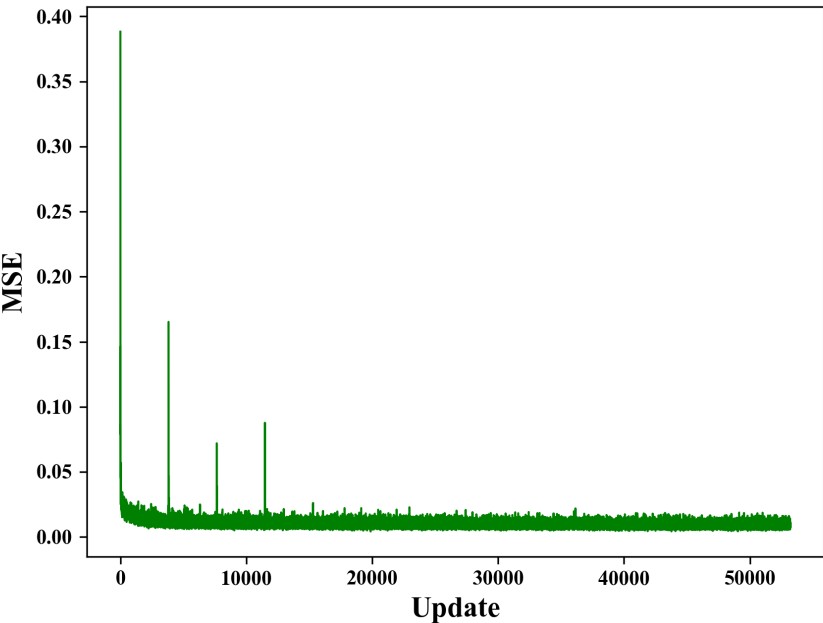

Figure 6: Training loss of the value estimator for the logic synthesis problem.

During testing, the number of the candidate nodes is fixed at $K = 5$ in uniform sampling. In the clustering sampling strategy, $N_c = 5$ clusters are employed and 2 nodes are sampled from each cluster. The parameter $\eta$ is set to 0.2. In the UCT-like sampling, $K = 5$ and $c_b = 1.38$.

## K    An example of different search algorithms to solve logic synthesis

The search trees for A* search, MCTS, and SeeA* when solving the logic synthesis problem for the alu4 circuit are depicted in Figure 7, 8, and 9, respectively. For A* search, the initial three actions are consistently "113", indicating that the search process is trapped in a particular branch due to the lack of exploration. Due to the enforced exploratory nature, MCTS expands nodes across excessive branches, which can impede the efficiency of the search algorithm in generating solutions. SeeA* expands a moderate number of branches, striking a balance between the concentrated exploration of A* and the excessive expansion across multiple branches in MCTS. This approach allows for exploratory behavior without being confined to a single branch, while the selection of the candidate node with the minimum $f$-value prevents excessive expansion into irrelevant branches. As a result, SeeA* exhibits significant improvements in efficiency compared to both MCTS and A*.

## L    Value estimator for Sokoban problem

Sokoban is a puzzle video game where the objective is to move the crates strategically to push each crate to its corresponding storage locations. The input of the neural network is a four-dimensional tensor representing the positions of the box, the target, the person, and the walls, respectively. The architecture of the value estimator is summarized as:

- A convolutional layer with a kernel size of $64 \times 3 \times 3$.
- A batch normalization layer.

- A ReLU activation layer.

- A ResNet with three residual blocks.

- A convolutional layer with a kernel size of $1 \times 1 \times 1$.

- A batch normalization layer.

- A ReLU activation layer.

- A fully connected layer with dimensions $[100, 1]$.

- A ReLU activation layer.

The DeepCubeA paper provides $50,000$ training Sokoban problems and $1,000$ testing Sokoban problems. The A$^*$ search guided by a manually designed heuristic is employed to find solutions for the training problems. $g$ is the number of steps arriving at the current state. $h$ is the sum of distances between the boxes and their respective goals, as well as the distance between the person and the nearest box. Under limited search time, $46,252$ training problems are solved. For each collected trajectory $\{n_0^i, n_1^i, \cdots, n_t^i, \cdots, n_{T_i}^i\}$, the learning target for state $n_t^i$ is the number of steps from $n_t^i$ to the goal state $n_{T_i}$:

$$z(n_t^i) = T_i - t. \tag{59}$$

Mean square error is employed as the loss function:

$$L(\theta) = \frac{\sum_i \sum_t (v(n_t^i; \theta) - z(n_t^i))^2}{\sum_i T_i}. \tag{60}$$

Adam optimizer with a $0.0001$ learning rate is used to update the parameters.

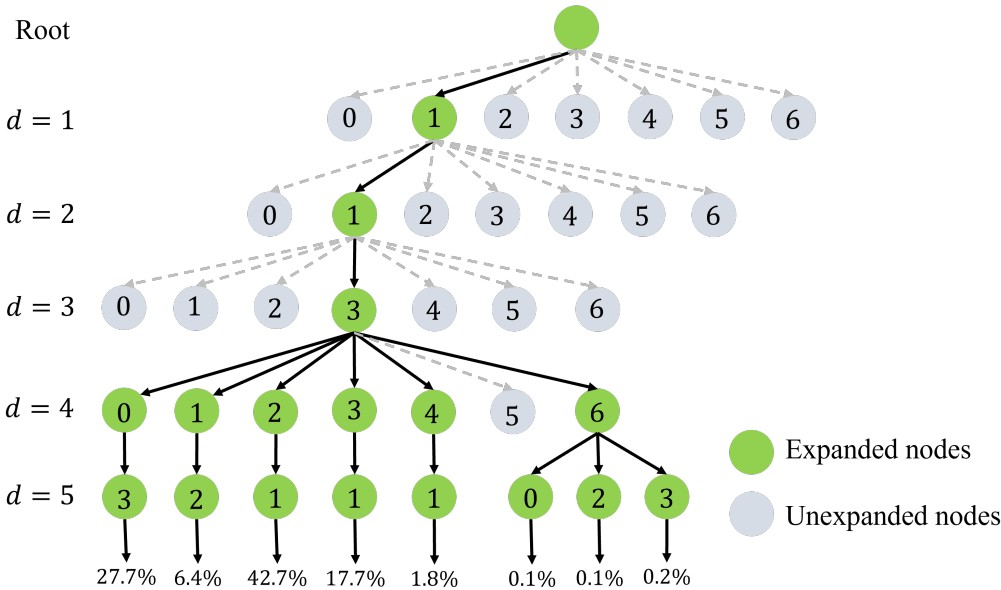

Figure 7: The search tree of A$^*$ search when solving logic synthesis problem for alu4.

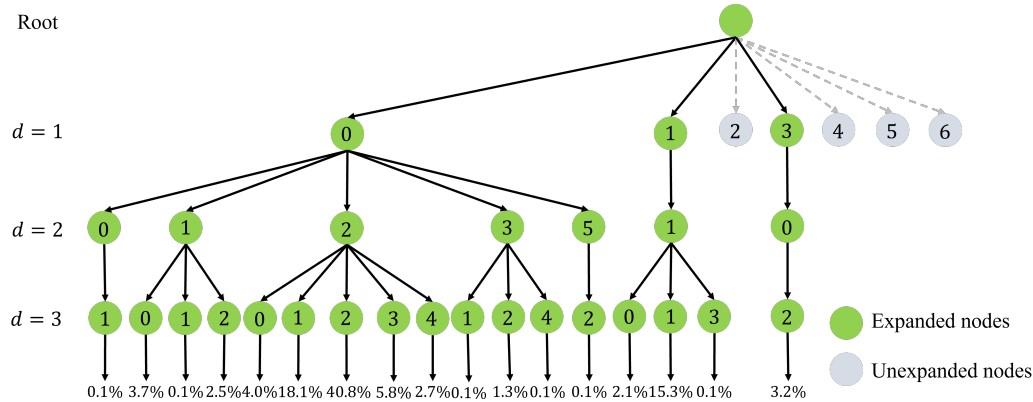

Figure 8: The search tree of MCTS when solving logic synthesis problem for alu4.

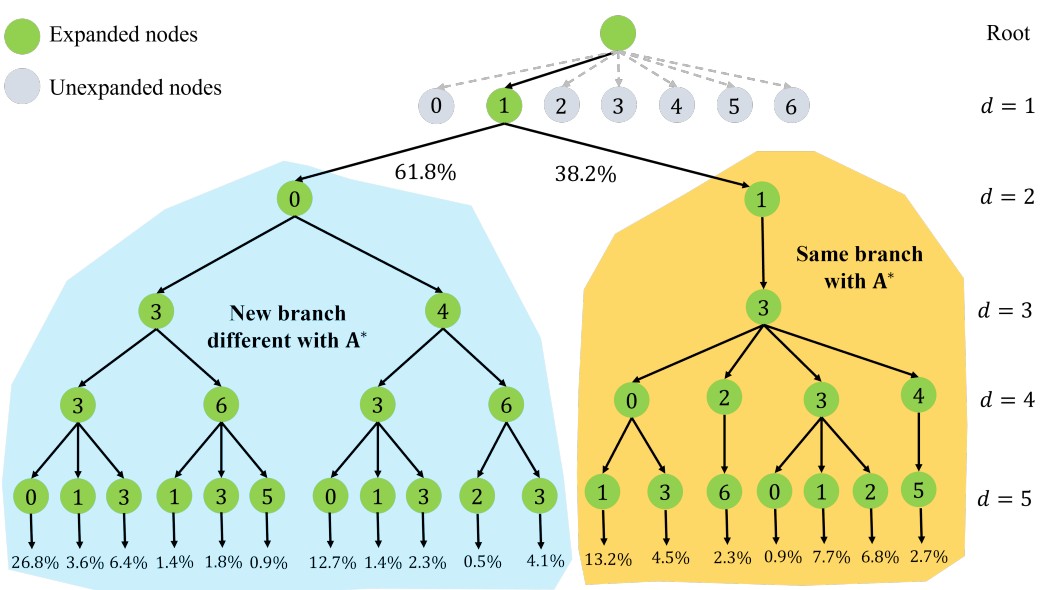

Figure 9: The search tree of SeeA* search when solving logic synthesis problem for alu4.

## M  Test results for Sokoban problem

The search process allows a maximum of $10^5$ expansions. The size of the candidate set in SeeA* is set to $K = 100$. In the clustering sampling strategy, $N_c = 2$ clusters are employed and the parameter $\eta$ is set to 0.01. In the UCT-like sample, $c_b$ is set 0.3. Experiment results are summarized in Table 7. With the exploration behavior induced by the selective sampling, SeeA* yields shorter solutions than $A^*$ with a slight increase in the number of expanded nodes. By setting $\varepsilon = 1.5$, WA* identifies feasible solutions using the minimum average number of node expansions. However, the solutions by WA* tend to have longer lengths, indicating that excessive reliance on the heuristic function biases the search towards suboptimal solutions.

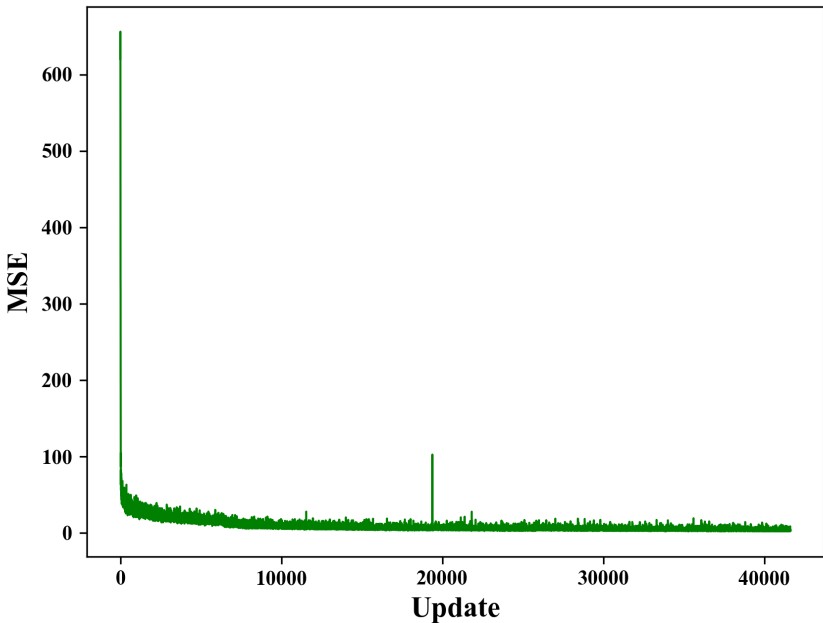

Figure 10: Training loss of the value estimator for the Sokoban problem.

Table 7: Test results on 1000 Sokoban cases. (Results of LevinTS, PHS, and DeepCubeA are provided by [38], [39], and [1], respectively.

| Algorithm | Solved | Length | Expansions |
|---|---|---|---|
| $A^*$ | **100.0**% | 32.664 | 1010 |
| $WA^*(\varepsilon = 1.5)$ | **100.0**% | 34.227 | **655** |
| LevinTS | **100.0**% | 39.8 | 6603 |
| PHS | **100.0**% | 39.1 | 2130 |
| DeepCubeA | **100.0**% | 32.9 | 1050 |
| $SeeA^*$(Uniform) | **100.0**% | 32.652 | 1069 |
| $SeeA^*$(Cluster) | **100.0**% | 32.689 | 1130 |
| $SeeA^*$(UCT) | **100.0**% | **32.505** | 1944 |

# N  Test results for path finding

To illustrate the effectiveness of SeeA$^*$ on problems where accurate heuristics could exist but the guiding heuristic used is unreliable, experiments on path finding are conducted, which is to find the shortest path from a starting point to a destination. The cost for each step is $1$. $g$ is the number of steps taken to reach the current position, and $h$ is the Euclidean distance from the current position to the target position, which is reliable enough to guide the A$^*$ search in early studies. $100$ robotic motion planning problems [3] are used to test the performance of A$^*$ and SeeA$^*$. Under the guidance of the same reliable $h$, both A$^*$ and SeeA$^*$ find the optimal solutions for all testing cases, for which the average length is $400$. To validate the superiority of SeeA$^*$, an unreliable heuristic function $\hat{h}$ is designed, which is randomly sampled from $[0, 2 \times h]$. During the search process, nodes are evaluated by $\hat{f} = g + \hat{h}$. In this situation, the average solution length of A$^*$ is $691.1$, much longer than SeeA$^*$'s $531.2$. Therefore, guided by an unreliable heuristic, SeeA$^*$ finds a better solution than A$^*$ search, demonstrating the superiority of SeeA$^*$.

# O   Investigations on the hyperparameters

Success rate and average solution length on the USPTO benchmark for the retrosyhthesis planning problem with different hyperparameter settings are displayed in Figure 11, 12 & 13.

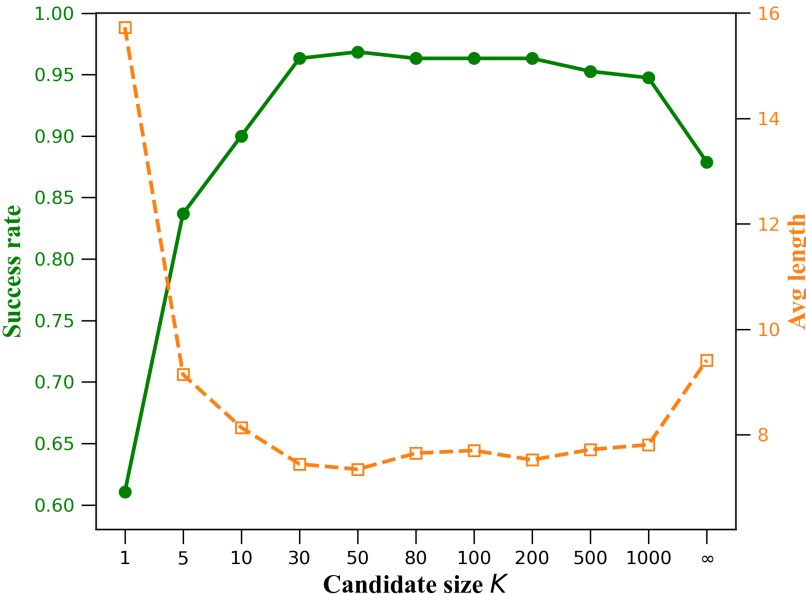

Figure 11: Success rate and average solution length on the USPTO benchmark with different candidate set sizes $K$ in uniform sampling strategy.

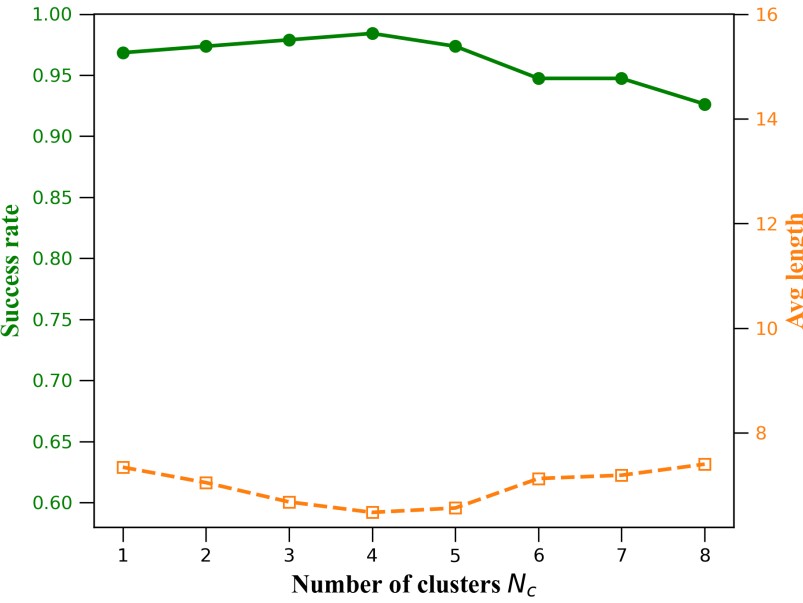

Figure 12: Success rate and average solution length on the USPTO benchmark with different number of clusters in clustering sampling strategy ($K = 50$).

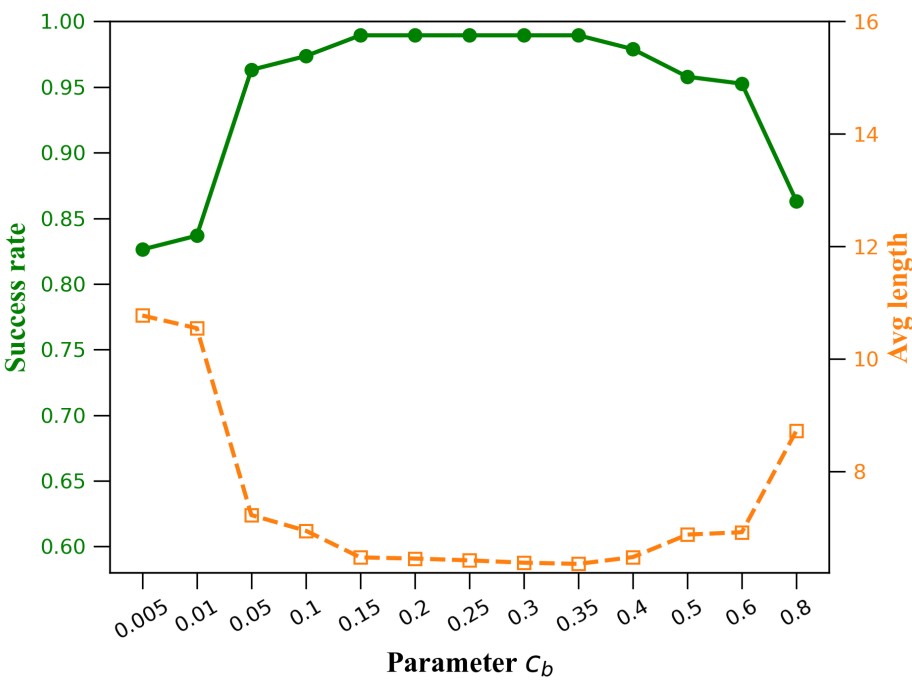

Figure 13: Success rate and average solution length on the USPTO benchmark with different $c_b$ in UCT-like sampling strategy ($K = 50$).

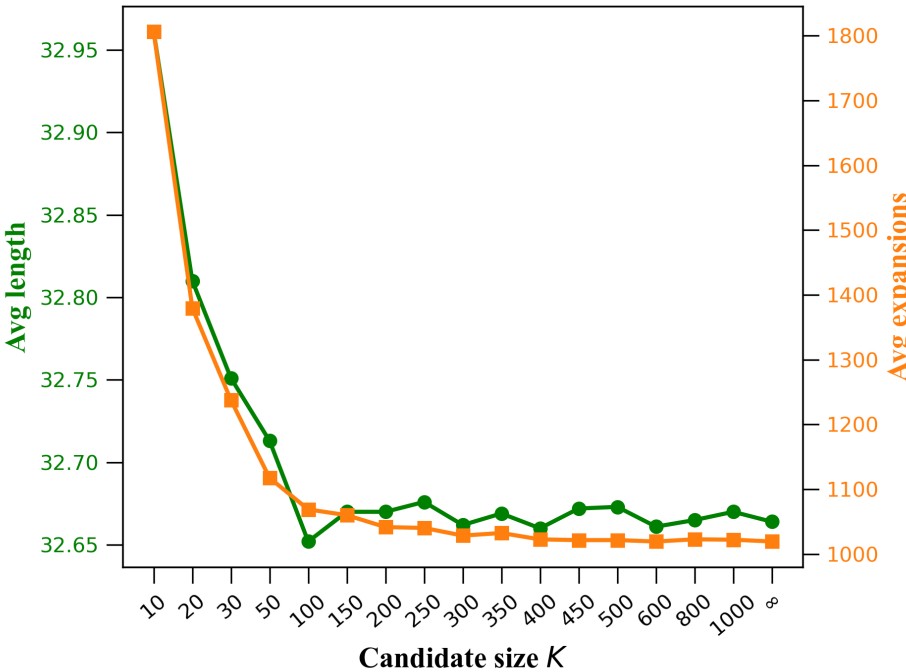

Figure 14: Average solution length and average number of node expansions tested on the Sokoban game with different candidate set sizes $K$ in uniform sampling strategy.

Ablation studies on logic synthesis are summarized below. The performance for different candidate set sizes $K$ for SeeA$^*$ with uniform sampling is displayed in Table 8. The performance is robust against different $K$, outperforming A* ($K = \infty$) consistently.

Table 8: Test results on logic synthesis with different $K$ in uniform sampling strategy.

| $K$ | ADP reduction rate | $K$ | ADP reduction rate |
|---|---|---|---|
| 1 | 19.8% | 20 | 21.2 |
| 3 | 22.1% | 30 | 19.7 |
| 5 | 21.6% | 50 | 19.8 |
| 10 | 19.8% | $\infty$ | 19.5 |

Table 9: Test results on logic synthesis with different $c_b$ in UCT-like sampling strategy.

| $c_b$ | ADP reduction rate | $c_b$ | ADP reduction rate |
|---|---|---|---|
| 0.5 | 20.8% | 1.38 | 22.5% |
| 1.0 | 21.8% | 1.5 | 22.6% |

The performance for different $c_b$ for UCT-like sampling is presented in Table 9, which is robust against different $c_b$. Enhanced exploration with a larger $c_b$ leads to superior performance and longer running time.

For the sokoban game, the average solution length and the average number of node expansions for uniform sampling strategy with different candidate sizes $K$ are presented in Figure 14. With the exploration behavior induced by the selective sampling, SeeA$^*$ yields shorter solutions than A$^*$ with a slight increase in the number of expanded nodes. The success rate of problem-solving is 100.0%. The ablation studies on the UCT-like sampling strategy are displayed in Table 10. Under a limited number of expansions, the stronger the exploration with a larger $c_b$, the shorter the identified solution path length, and the greater the number of expansions required to find a feasible solution. Sokoban permits at most four legal actions at each step, which makes the number of open nodes $N_o$ grow slowly. What's more, the state space of Sokoban ($10 \times 10$) is limited, and training a reliable cost function is relatively easier compared to retrosynthetic planning. According to Theorem 4.3, a larger $N_o$ and a less accurate cost estimator make the advantage of SeeA$^*$ more evident. Therefore, SeeA$^*$ is only slightly better than A$^*$ in Sokoban problem.

Table 10: Test results on Sokoban game with different $c_b$ in UCT-like sampling strategy ($K = 100$).

| $c_b$ | Solved | Length | Expansions |
|---|---|---|---|
| 0.00 | 100.0% | 32.664 | 1010 |
| 0.01 | 100.0% | 32.663 | 1026 |
| 0.05 | 100.0% | 32.660 | 1083 |
| 0.10 | 100.0% | 32.645 | 1219 |
| 0.15 | 100.0% | 32.625 | 1370 |
| 0.20 | 99.9% | 32.534 | 1451 |
| 0.25 | 99.9% | 32.499 | 1637 |
| 0.30 | 100.0% | 32.505 | 1944 |
| 0.35 | 99.7% | 32.366 | 1935 |
| 0.40 | 99.6% | 32.332 | 2138 |
| 0.45 | 99.6% | 32.316 | 2409 |
| 0.50 | 99.6% | 32.309 | 2751 |
| 0.70 | 98.9% | 32.030 | 3707 |

## P  Comparison between $\varepsilon$-Greedy and SeeA$^*$

Both $\varepsilon$-Greedy and SeeA$^*$ involve introducing exploration to A$^*$ search. $\varepsilon$-Greedy selects the node with the best $f$-value with a probability of $1 - \varepsilon$, and with a probability of $\varepsilon$, it randomly selects

a node from the remaining nodes. SeeA* with uniform sampling strategy selects a candidate set uniformly and the node with the best $f$ value within this candidate set is expanded. An example of the comparison of these two algorithms is given in Figure 15. There are five nodes $\{s_1, s_2, s_3, s_4, s_5\}$, and the $f$ values estimated by the heuristic value function are in increasing order. In A* search, node $s_1$ is expanded with a probability of 100%. In $\varepsilon$-Greedy ($\varepsilon = 0.5$), there is a 60% probability of expanding node $s_1$, and each of the remaining nodes has a 10% probability of being expanded. SeeA* expands $s_1$ (the node with the smallest $f$ value) with a probability of 60%. It expands $s_2$ (the node with the second smallest $f$ value) with a probability of 30%, and $s_3$ (the node with the third smallest $f$ value) with a probability of 10%. The remaining two nodes with the largest $f$ values are not eligible for expansion. Even though the selection of candidate nodes follows uniform sampling, which is quite random, only the candidate node with the smallest $f$ value is expanded. This guarantees the quality of the expanded nodes, as nodes with significantly worse $f$ values do not have a chance to be expanded, even if they might have been selected as candidates. Assuming there are $N_o$ open nodes and the size of the candidate set is $K$.

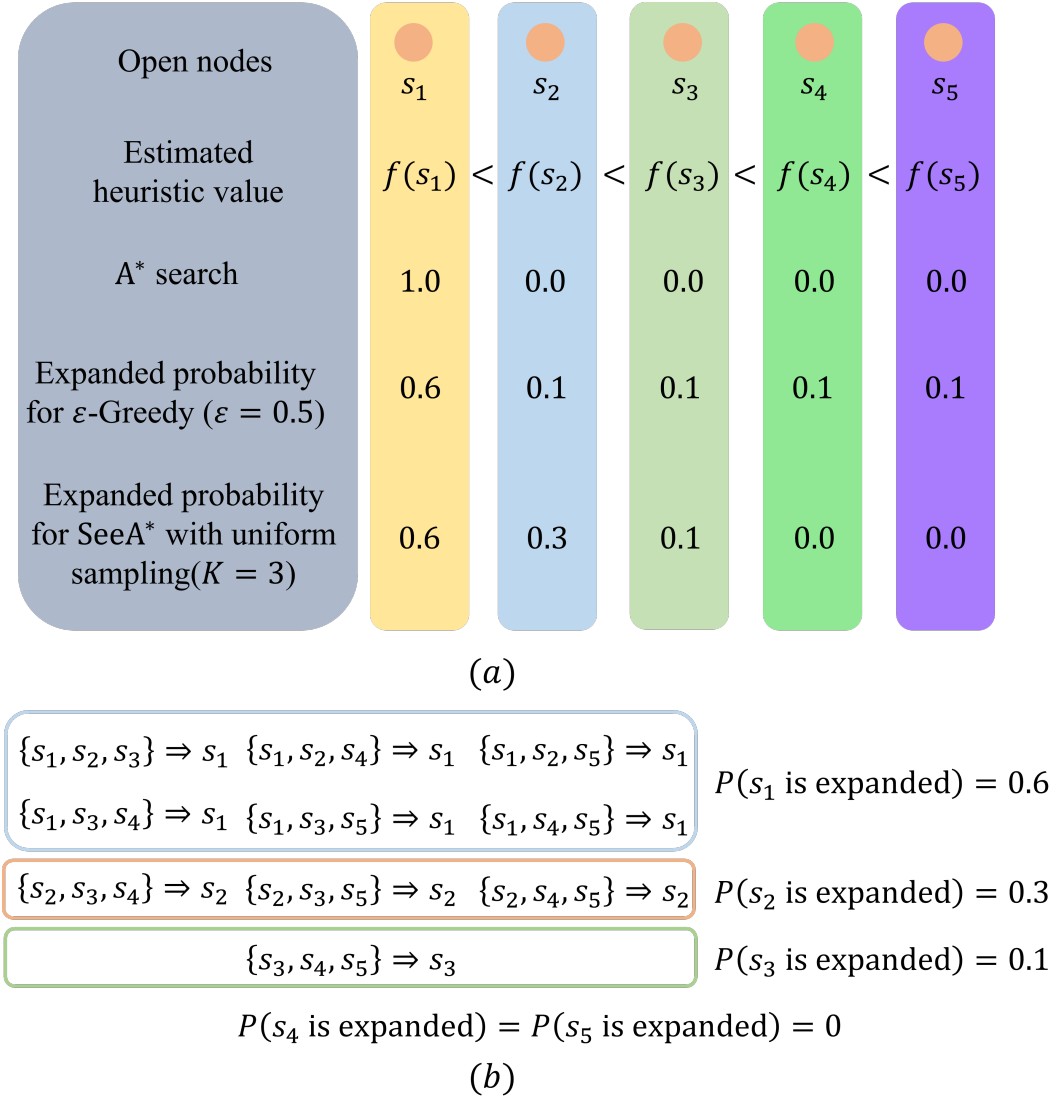

Figure 15: (a) An example of the difference of expansion probability for each node for $\varepsilon$-Greedy and SeeA* (Uniform sampling strategy is employed and the candidate size $K = 3$). (b) Expanded probability calculation for SeeA*. There are ten possible combinations of the candidate sets, and each combination occurs with equal probability.

The probability of expanding the node with the $n$th smallest $f$ value under the uniform sampling strategy can be derived as follows:

$$P(x^{(n)}) = \begin{cases} \frac{\binom{N_o-n}{K-1}}{\binom{N_o}{K}} = \frac{K(N_o-n)!(N_o-K)!}{N_o!(N_o-K-n+1)!}, & \text{if } n \leq N_o - K + 1 \\ 0, & \text{Otherwise} \end{cases} \tag{61}$$

The probability of being selected for expansion decreases gradually as the $f$ value increases. Compared to $\varepsilon$-Greedy, SeeA$^*$ utilizes the $f$ value to avoid excessive exploration in its decision process. Even though the estimated $f$ value may be not reliable enough to guarantee that the node with the smallest $f$ value is the real optimal node to be expanded, the $f$ value still can serve as a measure of node quality within a certain range. The heuristic function possesses a certain level of reliability, and it is generally inappropriate to either completely trust, like A$^*$ search, or completely disregard, like $\varepsilon$-Greedy, the evaluations provided by the heuristic function. SeeA$^*$ aims to achieve a balance between fully trusting and fully disregarding the evaluations of the heuristic function.

