# OpenReview forum: "SeeA*: Efficient Exploration-Enhanced A* Search by Selective Sampling"
_NeurIPS.cc/2024/Conference — NeurIPS 2024 oral_

### Official Review · Reviewer_KUFf · 2024-07-09

**Soundness:** 3
**Presentation:** 2
**Contribution:** 3
**Rating:** 7
**Confidence:** 3

**Summary:**

This paper proposes a novel search strategy, SeeA*. SeeA* employs a selective sampling process to screen a dynamic candidate subset based on an additional strategy. Under certain assumptions, SeeA* theoretically has better efficiency than A* search when SeeA* uses uniform select strategy and the heuristic value function deviates substantially from the true state value function. In experiments, SeeA* outperforms state-of-the-art heuristic search algorithms in terms of problem-solving success rate and solution quality while maintaining a low level of node expansions.

**Strengths:**

- Under certain assumptions, SeeA* theoretically has better efficiency than A* search.

- Experiments on two diverse real-world applications in chemistry and circuit design, as well as one puzzle-solving game, demonstrate the efficiency of SeeA*.

**Weaknesses:**

- Lots of information is in the appendix rather than the main text. Some parts should be moved to the main text, such as the algorithm pseudocode and results of different parameters in section N.

- Since this paper is about a traditional search algorithm and the learning part is not very clear, it may not be suitable for NeurIPS.

**Questions:**

Line 116: The functions \(\tau\) and \(c\) should be defined.

Figure 1: I don't understand why this figure lists \(n\) with \(10^4\). Because \(n_{10^4}\) won't be explored by either algorithm. Naive A* will expand \(n_{200}\) at most.

Line 152: I would suggest moving the algorithm part into the main paper from the appendix.

Line 158: When using uniform sampling, \(K \to \infty\) means SeeA* = A*, and \(K \to 1\) means SeeA* = Random Search.

Equation 52: The equation contains \(N\) rather than \(N_o\), and what is lowercase \(k\)?

Line 190: PUCT should be pUCT, and \(Q\) is not defined. What are the estimated \(f\) values? Since \(f = g + h\) which could be calculated directly, why do we need to estimate it?

Lines 201-203: I don't understand, is \(f^*\) sampled from Gaussian and \(f^*\) also sampled from Uniform?

Line 215: \(N_o\) is not defined.

Equation 8: We can get an optimal \(K = -1/\ln p\). For large prediction error, \(K \to 1\), random selection. For small prediction error, \(K \to \infty\), A*.

Table 1: Why is uniform slower than cluster as Uniform has smaller expansions, and cluster needs additional cluster process time?
Tables 1 and 2 should indicate whether lower or higher values are better.

Lines 14-16: "Theoretically establish the superior efficiency of SeeA* over A*" should mention the theoretical analysis based on specific assumptions.

And I'm also expecting an experiment on path search since this method is highly related to A*.

**Limitations:**

The strategy parameters are still hand-crafted, which may result in less dynamism when the test case distribution changes.

---

> ### Author Rebuttal · Authors · 2024-08-07
>
> Thank you for your valuable feedback. We will revise the paper accordingly and define symbols clearly. The key algorithms will be moved to the main paper. We hope that our response addresses your concerns.
>
> Q1: Why Figure 1 lists (n) with ($10^4$).
>
> A1: Setting the number of nodes to $10^4$ in the non-optimal branch is intended to demonstrate that even if the path is quite poor ($f=10^4<<f^*$), A* still spends significant efforts expanding nodes along that branch.
>
> Q2: Is ($f^*$) sampled from Gaussian and ($f^*$) also sampled from Uniform?
>
> A2: In our assumptions, the $f^*$ value is the ground truth evaluation, and $f$ aims to accurately predict $f^*$. The prediction error of $f$ with respect to $f^*$ follows a uniform distribution. The distribution of $f^*$ values for all non-optimal nodes is Gaussian. We will revise the original description to clarify the expression.
>
> Q3: Why is uniform slower than cluster as Uniform has smaller expansions, and cluster needs additional cluster process time?
>
> A3: There are primarily three parts. The first part is that competitive learning reduces the time required for clustering. One simply needs to assign each new node to the nearest center without an iterative process or re-clustering. The second part is that the time required for expanding each node varies. In retrosynthesis planning, expanding a node needs to utilize the RDKit package to reconstruct potential chemical reactions from the top $50$ chemical reaction templates selected based on the policy network. The number of chemical reactions corresponding to each reaction template varies, resulting in differences in computation time. The third part is the penalty for unresolved testing samples. If a test sample fails to identify a feasible solution, the expansion count for that sample is set to $500$, and the runtime is $600$ seconds, when calculating the mean performance listed in the table. The penalty for runtime is more severe than for expansion times. Due to the higher success rate of the Cluster sampling compared to the Uniform sampling, the discrepancy between running time and the number of expansions is possible.
>
> Q4: Experiment on path search since this method is highly related to A*.
>
> A4: Experiments on pathfinding problems are conducted using an existing heuristic function $h$ for guidance. The pathfinding problem is to find the shortest collision-free path from a starting point to a destination. The cost for each step is $1$. $g$ is the number of steps taken to reach the current position, and heuristic $h$ is the Euclidean distance from the current position to the target position, which is reliable to guide the A* search. The $100$ robotic motion planning problems [1] are used to test the performance. Under the guidance of the same reliable $h$, both A* and SeeA* find the optimal solution for all testing cases, and the average length is $400$. The number of expansions of SeeA*($K=5$) with uniform sampling is $33283.21$, slightly less than the $33340.52$ of A*. To validate the superiority of SeeA*, an unreliable heuristic function $\hat{h}$ is employed, which is randomly sampled from $[0, 2\times h]$. During the search process of A*, the node with the smallest $\hat{f}=g+\hat{h}$ is expanded. In this situation, the average solution length of A* is $691.1$, much longer than SeeA*'s $438.4$. Moreover, A* requires $50281.28$ expansions, which is significantly more than the $32847.26$ expansions needed by SeeA*. Therefore, guided by an unreliable heuristic function, SeeA* finds a better solution than A* with fewer expansions, demonstrating the superiority of SeeA*.
>
> [1] Bhardwaj, Mohak, Sanjiban Choudhury, and Sebastian Scherer. "Learning heuristic search via imitation." Conference on Robot Learning 2017.
>
> Q5: This paper is about a traditional search algorithm and the learning part is not very clear.
>
> A5: We believe that our paper is suitable for NeurIPS. MCTS with help of deep learning contributed crucially the success of AlphaZero. Deep learning is also able to contribute renaissance of A*, and three possible aspects are addressed with a family of possible improvements [2]. The first and also straightforward aspect is estimating $f$ with help of deep learning, which makes current studies on A* including this paper into the era of learning aided A*. The second aspect is seeking better estimation of $f$, such as scouting before expanding the current node to collect future information to revise $f$ of the current node, which takes a crucial rule for the success of AlphaGo. The third aspect is about selecting nodes among the OPEN list that consists of open nodes of A*. It is an old tune even in the classical era of A*, but investigation is seldomly made. As shown in Appendix D, J, and L, fully connected network, convolutional network, and graph network are utilized to estimate the $h$ function.
>
> Moreover, the part of sampling strategies is related to learning distributions of open nodes. Uniform sampling approximates the distribution based on frequencies. Clustering sampling is akin to use Gaussian mixture model to learn the distribution of open nodes, where each cluster is a Gaussian. Competitive learning is adopted to assigns nodes to save computing resources. The candidate nodes are sampled from the learned distribution.
>
> What's more, several papers focusing on enhancements to search algorithms have been published in the past NeurIPS conferences [3,4,5,6].
>
> [2] Xu, Lei. "Deep bidirectional intelligence: AlphaZero, deep IA-search, deep IA-infer, and TPC causal learning." Applied Informatics 2018.
>
> [3] Orseau, Laurent, et al. "Single-agent policy tree search with guarantees." NeurIPS 2018.
>
> [4] Sokota, Samuel, et al. "Monte carlo tree search with iteratively refining state abstractions." NeurIPS 2021.
>
> [5] Xiao, Chenjun, et al. "Maximum entropy monte-carlo planning." NeurIPS 2019.
>
> [6] Painter, Michael, et al. "Monte carlo tree search with boltzmann exploration." NeurIPS 2023.

---

> ### Author Response · Authors · 2024-08-07
> **Explanations of some paper symbol definitions**
>
> Q6: What are the estimated (f) values? Since (f = g + h) which could be calculated directly, why do we need to estimate it?
>
> A6: $f(n)$ is the evaluation of a node $n$, which is calculated by $g(n)+h(n)$. $g(n)$ is the accumulated cost from the starting node to $n$, which is obtained during the interaction process. $h(n)$ is the expected future cost from $n$ to the termination, which is unknown and needs to be estimated by a heuristic function. We will revise this sentence to avoid potential misunderstandings.
>
> Q7: Line 116: The functions ($\mathcal{T}$) and ($c$) should be defined.
>
> A7: $\mathcal{T}$ is the state transition function defined in a Markov decision process, which is used to obtain the following state $s_{t+1}$ when taking action $a_t$ at state $s_t$. $c$ is the cost function, which gives the received cost when taking action $a_t$ at state $s_t$. We will revise the paper to provide precise definitions.
>
> Q8 Line 190: PUCT should be pUCT, and (Q) is not defined.
>
> A8: pUCT is the summation of Q and U, where Q is the average value of the child nodes, and U is the exploration bonus. The definition of Q will be provided in more detail in the revised paper.
>
> Q9: Line 215: $N_o$ is not defined.
>
> A9: $N_o$ is the number of nodes in the open set $\mathcal{O}$. The definition of $N_o$ will be included in the revised paper.
>
> Q10: "Theoretically establish the superior efficiency of SeeA* over A*" should mention the theoretical analysis based on specific assumptions.
>
> A10: Thank you for pointing that out. We will revise our paper accordingly.
>
> Q11: N and k in Equation 52.
>
> A11: Thank you for pointing that out. $N$ and $k$ should correspond to $N_o$ and $K$ in the main text. We will revise Equation 52 to maintain the consistency in the symbols used.
>
> Q12: Tables 1 and 2 should indicate whether lower or higher values are better.
>
> A12: Thank you for your suggestion. We will indicate in Table 1 and 2 whether larger values are preferred or smaller values are preferred.
>
> Q13: When using uniform sampling, $K \to \infty$ means SeeA* = A*, and $K \to 1$ means SeeA* = Random Search.
>
> A13: We agree with your viewpoint. When $K \to \infty$, all open nodes are selected as the candidate nodes, and SeeA* degenerates back to A*. The expanded node is chosen using a best-first approach, relying entirely on the exploitation of the $f$ function. When $K \to 1$, only one node is selected as the candidate node, and hence, it is guaranteed to be expanded. The expanded node is determined by the exploration of the sampling strategy. Therefore, an appropriate value of $K$ ensures that SeeA* is a combination of A* search and random search, achieving a balance between exploitation and exploration.
>
> Q14: Equation 8: We can get an optimal ($K = -1/\log p$). For large prediction error, ($K \to 1$), random selection. For small prediction error, ($K \to \infty$), A*.
>
> A14: We agree with your viewpoint. Equation 8 reaches its maximum value when $K=-1/\log p$, at which point SeeA performs optimally. For large prediction errors, $p$ approaches $0$, and SeeA* degrades to random selection by setting $K=1$. For small prediction errors, $p$ approaches $1$, and SeeA* is the same as A* by setting $K=\infty$. Intuitively, if the prediction error is sufficiently small, then the best-first A* search is optimal. If the prediction error is significantly large, decisions based on $f$ values are likely to be misleading, and random sampling may perform better in this case. We will add this analysis to the main text. Thank you for your suggestion.
>
> Q15: The strategy parameters are still hand-crafted, which may result in less dynamism when the test case distribution changes.
>
> A15: At present, the hyperparameters of algorithms indeed require manual design, but the performance is robust against different hyperparameter settings. The automated design to adapt to dynamically changing environments deserves more research efforts in the future.

---

> > ### Comment · Reviewer_KUFf · 2024-08-14
> >
> > Thank you for the responses to all my questions. I have no other question.

---

> > > ### Author Response · Authors · 2024-08-14
> > >
> > > Dear Reviewer KUFf:
> > >
> > > Thank you very much for your response! We are glad to hear that your concerns have been addressed. The feedback is very helpful, and we will revise our next version of the paper accordingly. Thank you.
> > >
> > > Best regards！

---

### Official Review · Reviewer_BBaT · 2024-07-13

**Soundness:** 3
**Presentation:** 3
**Contribution:** 3
**Rating:** 6
**Confidence:** 4

**Summary:**

The paper introduces a method for prioritizing nodes for expansion during heuristic search that builds on A* search. However, instead of selecting the node with the lowest cost in OPEN, it samples a subset of OPEN and selects the node with the lowest cost from that subset. The sampling procedure is done using a uniform, clustering, and UCT approach. Results show that this sample based approach performs better than regular A* search. Perhaps surprising, this includes the uniform sampling method.

**Strengths:**

The paper gives a good motivation for when sampling a subset of the OPEN list may be advantageous. In specific, this is when the heuristic function contains significant inaccuracies. The results also show the consistent superiority of this approach over A* search.

**Weaknesses:**

I wonder what the results would be in an environment

Line 123: Step 3 also occurs if a node is associated with a state in CLOSED, but is find via a shorter path

**Questions:**

Is the clustering strategy susceptible to collapsing to a single cluster? Since the initialization is purely random, it seems like this could be the case. Is there any empirical evidence to suggest this does or does not happen?

How is the heuristic function for Sokoban trained?

It seems the uniform sampling method performs better than A* search. Is there any intuition as to why? For environments with large branching factors, where deep search trees are needed, and where shortest paths are sparse, the probability of sampling a subset of OPEN that contains nodes on a shortest path would be small. Would this not hurt performance for uniform?

**Limitations:**

In the questions section, I describe a scenario in which random sub sampling of OPEN could hurt performance. I am not sure if this is definitely the case, but, if it is, then I would consider that a limitation.

---

> ### Author Rebuttal · Authors · 2024-08-07
>
> Thank you very much for your valuable feedback. We hope that our response addresses any concerns you may have.
>
> Q1:Is the clustering strategy susceptible to collapsing to a single cluster?
>
> A1: This scenario may indeed occur, and in such situations, running multiple iterations with varied initializations can be considered. Developing more efficient sampling strategies is a crucial future endeavor. In practical applications outlined in the paper, this scenario occurs infrequently in general. It occurs more often when the number of clusters is relatively low, but it may be justifiable to obtain only one cluster in some cases with fewer nodes in the search tree. For example, when the number of clusters is initialized at $5$, in the retrosynthesis task, all the generated nodes of $74$ (out of $190$) testing samples are assigned to the same one cluster after the search is terminated. The average number of node expansions to identify a feasible solution is $79.75$. When we look into the details of the $74$ samples, $53$ ones have found a feasible solution within less than $20$ expansions, and $72$ samples have found a feasible solution within less than $79.75$ expansions. What's more, increasing the number of initial cluster centers and then selecting only clusters containing nodes evenly can be a potential solution to avoid the clustering strategy collapsing to a single cluster. For the above mentioned retrosynthesis task, when the number of initial cluster centers is $20$, the number of samples with only one cluster decreases from $74$ to $41$.
>
> Q2: How is the heuristic function for Sokoban trained?
>
> A2: Training details will be added to the appendix. The DeepCubeA paper provides $50,000$ training Sokoban problems and $1,000$ testing Sokoban problems. The A* search guided by a manually designed heuristic is employed to find solutions for the training problems. $g$ is the number of steps arriving at the current state. $h$ is the sum of distances between the boxes and their destinations, along with the distance between the player and the nearest box. Under limited search time, $46,252$ training problems are solved. For each collected trajectory  $\{n_0^i,n_1^i,\cdots,n_t^i,\cdots,n_{T_i}^i\}$, the learning target for state $n_t^i$ is the number of steps from $n_t^i$ to the goal state $n_{T_i}$:
> $$z(n_t^i)=T_i-t.$$
> Mean square error is employed as the loss function:
> $$L(\theta)=\frac{\sum_{i}\sum_{t}(v(n_t^i;\theta)-z(n_t^i))^2}{\sum_i T_i}$$
> Adam optimizer with a $0.0001$ learning rate is used to update the parameters.
>
> Q3: It seems the uniform sampling method performs better than A* search. Is there any intuition as to why? For environments with large branching factors, where deep search trees are needed, and where shortest paths are sparse, the probability of sampling a subset of OPEN that contains nodes on a shortest path would be small. Would this not hurt performance for uniform?
>
> A3: SeeA* uniformly samples candidate nodes and expands the one with the lowest $f$ value. In this setting, each nodes, except a few with the worst $f$ values, has a probability of being expanded, and the node with a smaller $f$ value still has a larger expansion likelihood, as discussed in Appendix O. SeeA* improves exploration compared to A*, but expansion remains mainly concentrated on nodes with the best $f$ values. Therefore, SeeA* achieves a balance between exploration and exploitation, making it better than A*.
>
> If the optimal node $n_1$ is expanded by A*, $f(n_1)$ must be less than the $f$ values of all $N_o$ open nodes, while SeeA* only needs to select $n_1$ from $K$ candidate nodes if $n_1$ is included in the candidate set $\mathcal{D}$.
> $$P_{A^*}(n_1 \text{ is expanded})=P(n_1=\arg\min_{n\in\mathcal{O}}f(n))$$
> $$P_{SeeA^*}(n_1 \text{ is expanded})=P(n_1=\arg\min_{n\in\mathcal{D}}f(n)|n_1\in\mathcal{D})P(n_1\in\mathcal{D})$$
> SeeA* outperforms A* if
> $$P(n_1\in\mathcal{D})>P(n_1=\arg\min_{n\in\mathcal{O}}f(n)) / P(n_1=\arg\min_{n\in\mathcal{D}}f(n))=p_{\sigma}^{N_o-1}/p_{\sigma}^{K-1}=p_{\sigma}^{N-k}.$$
>
> According to Corollary 4.2, the larger the prediction error $\sigma$, the lower the likelihood $p_{\sigma}$ that the $f(n_1)$ is smaller than the $f$ value of a non-optimal node. $P_O=P(n_1=\arg\min_{n\in\mathcal{O}}f(n))$ is the product of $N_o-1$ probabilities, while $P_D=P(n_1=\arg\min_{n\in\mathcal{D}}f(n))$ is the product of $K-1$ probabilities. When $\sigma$ increases, $P_O$ decreases much faster than $P_D$. The right side of the above inequality decreases. For uniform sampling, $P(n_1\in\mathcal{D})=K/N_o$ is irrelevant with $\sigma$. Therefore, even with uniform sampling, SeeA* can outperform A* when $\sigma$ are large enough.
>
> In scenarios with large branching factors, the probability of sampling a candidate set containing optimal nodes is lower. Taking uniform sampling as an example, $P(n_1\in\mathcal{D})=K/N_o$ decreases as $N_o$ increases. However, the probability of A* selecting the optimal node will also decrease significantly with $N_o$ because every node has a probability of better than $n_1$. As presented in Equation 10, $H(N_o)$ approaches $1$ as $N_o$ approaches infinity, ensuring the inequality in Theorem 4.3 holds. Despite the potential decline in SeeA*'s performance with the growth of $N_o$ inevitably, it continues to outperform A*.
>
> Q4: I wonder what the results would be in an environment Line 123: Step 3 also occurs if a node is associated with a state in CLOSED, but is find via a shorter path.
>
> A4: If a node is associated with a state in CLOSED but is found via a shorter path, this node is expanded as Line 123. Experiments are conducted in retrosynthesis planning. The performance is similar to the results in the paper, and SeeA* still outperforms A* with higher success rates and shorter solution lengths.
>
> |Algorithm|Solved|Length|Expansions|
> |-|-|-|-|
> |A*|88.42%|9.28|91.27|
> |SeeA*(Uniform)|96.32%|7.44|69.21|
> |SeeA*(Cluster)|96.84%|7.04| 64.84|
> |SeeA*(UCT)|98.95%|6.36|56.87|

---

### Official Review · Reviewer_8E9W · 2024-07-20

**Soundness:** 3
**Presentation:** 4
**Contribution:** 3
**Rating:** 7
**Confidence:** 4

**Summary:**

This work introduces a refined version of the A* search algorithm that integrates selective sampling to improve exploration & efficiency. The developed algorithm balances exploration and exploitation when heuristic guides are off the mark with the help of three sampling strategies. Also it outperforms traditional A* in both the quality of solutions and computational efficiency that is backed with practical tests.

**Strengths:**

1. Work uses multiple (3) sampling strategies to balance between exploration & exploitation for diverse scenarios
2. Good theoretical analysis to show that the developed algorithm is efficient than traditional A* (when heuristic functions deviate from true state values)
3. Validates the effectiveness of the algorithm through extensive experiments across multiple application
4. Algorithm can handle large & complex search spaces (backed by experiments)

**Weaknesses:**

1.	Limited theoretical analysis for some sampling strategies
2.	Strong assumptions in theoretical results
3.	Experiment represents a narrow spectrum
4.	Scalability on resource-constrained environments/ large-scale system is not discussed
5.	No sufficient details on when SeeA* fails (or) perform suboptimally

**Questions:**

-	In Alg 1, the termination condition “until O is empty” is not clear. For some challenging problems, if a solution may not exist, will SeeA* keep expanding nodes indefinitely? Consider adding a maximum iteration limit to guarantee termination?
-	The uniform sampling strategy discussed in Sec. 4.1.1 is clear and straightforward. However, the authors could discuss the potential drawbacks of this strategy, such as the possibility of selecting low-quality nodes, and how SeeA* addresses these drawbacks
-	The assumption in Corollary 4.2 that the prediction error for f* is uniformly distributed is quite strong. In practice, it’s more likely non-uniform like Gaussians? More info. on the sensitivity of the theoretical results to this assumption and empirical validation of the distribution of prediction errors will be helpful
-	NIT: The specific f and f* values seem arbitrary in Figure 1. Are these values generalize broadly (or) are they cherry-picked?
-	The authors could provide more intuition on the implications of Theorem 4.3. How does the result relate to the trade-off between exploration and exploitation in SeeA*, and how can it guide the selection of the sampling strategy or hyperparameters?
-	In Sec. 5.2, the authors compare SeeA* with various baselines, including MCTS [1]. However, it is mentioned that the MCTS in [1] did not utilize any guiding heuristics. It would be informative to compare SeeA* with an MCTS variant that uses the same guiding heuristics for a fair comparison
-	The hyperparameter sensitivity analysis in Sec. 5.4 provides insights on the performance of SeeA*. However, the analysis is limited to the retrosynthetic planning problem. Does this generalize to other two problem domains and beyond?
-	In the Sokoban experiment (Sec. 5.3), the authors compare SeeA* with several baselines, including DeepCubeA. However, the experimental setup for DeepCubeA is not clearly stated, making it difficult to asses the fairness of the comparison.

Reference:
[1] Walter Lau Neto, Yingjie Li, Pierre-Emmanuel Gaillardon, and Cunxi Yu. Flowtune: End-to-end automatic logic optimization exploration via domain-specific multi-armed bandit. IEEE Transactions on Computer-Aided Design of Integrated Circuits and Systems, 2022.

**Limitations:**

-	The experiment focusses solely on synthetic search problems. Initial results on real-world domain for practical applicability would be valuable
-	Paper provides theoretical comparison of SeeA* vs A* for uniform sampling strategy, but lacks analysis for other proposed sampling strategies.
-	Paper only tests on problems that don’t have inaccurate heuristics. Testing on additional domains that have unreliable heuristics would be a good addition

---

> ### Author Rebuttal · Authors · 2024-08-07
>
> We thank the reviewer for constructive comments and suggestions. We will revise our paper carefully. Hope our explanation below can address your concerns.
>
> Q1: Adding a maximum iteration limit to guarantee termination.
>
> A1: Adding a limit to guarantee termination is necessary, and is adopted in our experiments. In retrosynthesis planning, search algorithms are limited to a maximum of $500$ policy calls, or $10$ minutes of running time, as mentioned in Line 271. In Sokoban, the search process is terminated if the running time exceeds $10$ minutes. We will revise Alg 1 accordingly.
>
> Q2: The potential drawbacks of uniform sampling strategy.
>
> A2: The uniform sampling strategy is easy to implement, but $P(n_1\in\mathcal{D})$, the probability of selecting the optimal node $n_1$ to the candidate set $\mathcal{D}$, is relatively low, which directly impacts $P_S$, as shown in Equation 7. Therefore, additional strategies, clustering sampling and UCT-like sampling, are designed to improve $P(n_1\in\mathcal{D})$ by increasing the diversity of the selected nodes to avoid the excessive concentration of a few branches, thereby increasing $P(n_1\in\mathcal{D})$. If more information is available besides the $f$ evaluation, superior strategies can be designed, such as a specialized policy model.
>
> Q3: The assumption in Corollary 4.2 is quite strong. Prediction error is more likely non-uniform like Gaussians.
>
> A3: Thank you for your suggestion. We can prove that Corollary 4.2 also holds when noise follows a Gaussian distribution. Please refer to Global Rebuttal A1.
>
> Q4: Are f and f* values generalize broadly (or) are they cherry picked?
>
> A4: A specific example is provided in Figure 1 to illustrate that A* may be trapped in a local optimum due to insufficient exploration, which is not uncommon due to the prediction errors of guiding heuristics. Figure 7 & 9 in the appendix displayed the search tree of A* and SeeA* while solving a logic synthesis problem. A* exhibits an excessive concentration of expanded nodes in a particular non-optimal branch, which is the same as Figure 1. The superior performance of SeeA* over A* also corroborate this assertion.
>
> Q5: More intuition on the implications of Theorem 4.3.
>
> A5: Thanks for your suggestion. More detailed discussions will be added to the paper. Please refer to Global Rebuttal A2.
>
> Q6: It would be informative to compare SeeA* with an MCTS using the same guiding heuristics for a fair comparison.
>
> A6: In Table 2, the results of MCTS guided by the same heuristics as SeeA* are displayed in the PV-MCTS row, which achieves a 19.5% ADP reduction, surpassing MCTS's 18.5% but falling short of SeeA*'s 23.5%. We will further clarify the distinction between the two in the revised paper.
>
> Q7: The hyperparameter analysis is limited to the retrosynthetic planning problem. Does this generalize to other two problem domains and beyond?
>
> A7: Ablation studies on the Sokoban have been provided in the appendix. As presented in Figure 11, the performance of SeeA* is stable across a wide range of $K$. As shown in Table 8, the stronger the exploration, the shorter the identified solution path length, and the greater the number of expansions required to find a feasible solution. Due to the relatively low difficulty levels of the testing examples in the Sokoban, constructing an accurate value predictor is easier than with the other applications. Therefore, SeeA* is only slightly better than A*.
>
> Ablation studies on logic synthesis are summarized below. The performance for different candidate set sizes $K$ for SeeA* with uniform sampling is displayed. The performance is robust against different $K$, outperforming A* ($K=\infty$) consistently.
>
> |K|1|3|5|10|20|30|50|$\infty$|
> |-|-|-|-|-|-|-|-|-|
> |ADP reduction (%)|19.8|22.1|21.6|19.8|21.2|19.7|19.8|19.5|
>
> The performance for different $c_b$ for UCT-like sampling is as follows, which is robust against different $c_b$. enhanced exploration with a larger $c_b$ leads to superior performance and longer running time.
>
> |$c_b$|0.5|1.0|1.38|1.5|
> |-|-|-|-|-|
> |ADP reduction (%)|20.8|21.8|22.5|22.6|
>
> Q8: Paper only tests on problems that don’t have inaccurate heuristics. Testing on additional domains that have unreliable heuristics would be a good addition.
>
> A8: Thank you for your suggestion. In the paper, two applications where obtaining accurate heuristics is challenging are considered. To illustrate the effectiveness of SeeA* on problems where accurate heuristics could exist but the guiding heuristic used is unreliable, experiments on pathfinding are conducted, which is to find the shortest path from a starting point to a destination. The cost for each step is 1. $g$ is the number of steps taken to reach the current position, and $h$ is the Euclidean distance from the current position to the target position, which is reliable to guide the A* search. $100$ robotic motion planning problems [4] are used to test the performance of A* and SeeA*. Under the guidance of the same reliable $h$, both A* and SeeA* find the optimal solutions for all testing cases, for which the average length is $400$. The number of expansions of SeeA*($K=5$) with uniform sampling is $33283.21$, slightly less than the $33340.52$ of A*. To validate the superiority of SeeA*, an unreliable heuristic function $\hat{h}$ is employed, which is randomly sampled from $[0, 2\times h]$. During the search process, nodes are evaluated by $\hat{f}=g+\hat{h}$. In this situation, the average solution length of A* is $691.1$, much longer than SeeA*'s $438.4$. Moreover, A* requires $50281.28$ expansions, which is significantly more than the $32847.26$ expansions needed by SeeA*. Therefore, guided by an unreliable heuristic, SeeA* finds a better solution than A* with fewer expansions, demonstrating the superiority of SeeA*.
>
> [4] Bhardwaj, Mohak, Sanjiban Choudhury, and Sebastian Scherer. "Learning heuristic search via imitation." Conference on Robot Learning. PMLR, 2017.

---

> > ### Comment · Reviewer_8E9W · 2024-08-14
> >
> > Dear Authors,
> >
> > Thank you for the detailed responses to all my questions. The responses to Q1, Q3, Q7 and Q9 align directionally with my comments and questions, thus addressing some of my questions. Therefore, I have increased my rating. I hope the authors can add relevant context in this rebuttal in the next iteration of this work.

---

> > > ### Author Response · Authors · 2024-08-14
> > >
> > > Dear Reviewer 8E9W:
> > >
> > > Thanks for your feedback! It really helped improve the quality of the paper. We will make sure to include the added results and discussion in the next version of this work. Thank you.
> > >
> > > Best regards！

---

> ### Author Response · Authors · 2024-08-07
>
> Q9: The experimental setup for DeepCubeA is not clearly stated,.
>
> A9: The experimental setup DeepCubeA is the same as the original paper [5]. The number of nodes expanded at each step is $1$, and the weight of the heuristic value is $0.8$. The test samples for SeeA* are the same as those for DeepCubeA. More details will be added to the appendix materials.
>
> [5] Agostinelli, Forest, et al. "Solving the Rubik’s cube with deep reinforcement learning and search." Nature Machine Intelligence 1.8 (2019): 356-363.
>
> Q10: Paper provides theoretical comparison of SeeA* vs A* for uniform sampling strategy, but lacks analysis for other proposed sampling strategies.
>
> A10: For computational simplicity, only uniform sampling was considered in the theoretical portion. Based on Equation 7, the probability of expanding the optimal node $n_1$ by SeeA* is
> $$
> P_S(\sigma)=P(n_1\in\mathcal{D})P(n_1=\arg\min_{n'\in\mathcal{D}}f(n')|n_1\in\mathcal{D})
> $$
> $P(n_1\in\mathcal{D})=K/N_o$ for uniform sampling. The other two sampling strategies aim to achieve a higher $P(n_1\in\mathcal{D})$ compared to uniform sampling by constructing a more diverse candidate set, thereby enhancing the likelihood of expanding the optimal node. Uniform sampling approximates the distribution based on frequencies, while clustering sampling is akin to use Gaussian mixture model to learn the distribution of open nodes, where each cluster is a Gaussian. The candidate nodes are sampled from the learned distribution. We will provide the theoretical analysis of other sampling strategies in the future.

---

### Author Rebuttal · Authors · 2024-08-07

A1: The assumption in Corollary 4.2 that the prediction error for $f^*$ is uniformly distributed is quite strong. To further illustrate the applicability of the algorithm, we also prove that Corollary 4.2 is established if the noise follows a Gaussian distribution. Denoting Gaussian distribution as $\mathcal{G}(\cdot,\cdot)$, Assumption 4.1 will be:

_For each node $n$ on the optimal path, $f(n)\sim\mathcal{G}(\mu_0^f,\sigma^2)$. For nodes not on the optimal path, $f(n)\sim\mathcal{G}(f^*(n),\sigma^2)$, and ${f^*(n)}$ are independently and identically sampled from $\mathcal{G}(\mu_1^f,\sigma^2_s)$. $\mu_0^f< \mu_1^f$ holds because the optimal path has a lower cost._

For two Gaussian distributions, we have the following lemma [1]:

Lemma 1: Assume $x\sim\mathcal{G}(\mu_1,\sigma_1^2)$, $y\sim\mathcal{G}(\mu_2,\sigma_2^2)$. If $x$, $y$ are independent of each other and $\mu_2>\mu_1$, then_
$$P(x>y)=\frac{1}{\pi}\int_0^{\frac{\pi}{2}}\exp\left\\{-\frac{1}{2}\frac{[(\mu_2-\mu_1)/\sqrt{\sigma_1^2+\sigma_2^2}]^2}{\cos^2\theta}\right\\}d\theta.$$
For a node $n$ on the optimal path, $f(n)\sim\mathcal{G}(\mu_0^f,\sigma^2)$. For a node $n'$ off the optimal path, $f(n')\sim\mathcal{G}(f^*(n'),\sigma^2)$. If $\mu_0^f>f^*(n')$:
$$P(f(n)<f(n')|\mu_0^f>f^*(n'))=\frac{1}{\pi}\int_0^{\frac{\pi}{2}}\exp\left\\{-\frac{1}{2}\frac{(f^*(n')-\mu_0^f)^2}{2\sigma^2\cos^2\theta}\right\\}d\theta=m(f^*(n')|\sigma)$$
Otherwise:
$$P(f(n)<f(n')|\mu_0^f<f^*(n'))=1-\frac{1}{\pi}\int_0^{\frac{\pi}{2}}\exp\left\\{-\frac{1}{2}\frac{(f^*(n')-\mu_0^f)^2}{2\sigma^2\cos^2\theta}\right\\}d\theta=1-m(f^*(n')|\sigma)$$

$$F(\sigma)=P(f(n)<f(n')|\sigma)=\int_{f^*(n')<\mu_0^f}P(f^*(n'))m(f^*(n')|\sigma)df^*(n')+\int_{f^*(n')\geq\mu_0^f}P(f^*(n'))(1-m(f^*(n')|\sigma))df^*(n')$$
If $\sigma_2>\sigma_1$:
$$F(\sigma_2)-F(\sigma_1)=\int_{f^*(n')<\mu_0^f}P(f^*(n'))(m(f^*(n')|\sigma_2)-m(f^*(n')|\sigma_1))df^*(n')+\int_{f^*(n')\geq\mu_0^f}P(f^*(n'))(m(f^*(n')|\sigma_1) - m(f^*(n')|\sigma_2))df^*(n')$$
$m(f^*(n')|\sigma)$ is symmetric about the axis $f^*(n')=\mu_0^f$, $m(f^*(n')|\sigma)=m(2\mu_0^f-f^*(n')|\sigma)$.
$$F(\sigma_2)-F(\sigma_1)=\int_{f^*(n')\geq\mu_0^f}(P(2\mu_0^f-f^*(n')) - P(f^*(n')))(m(f^*(n')|\sigma_2) - m(f^*(n')|\sigma_1))df^*(n')$$
According to the definition, $m$ is monotonically increasing with respect to $\sigma$. Therefore, $m(f^*(n')|\sigma_2) - m(f^*(n')|\sigma_1)>0$. Because $f^*(n')\sim\mathcal{N}(\mu_1^f,\sigma_2^2)$ and $\mu_0^f<\mu_1^f$, we have $P(2\mu_0^f-f^*(n'))-P(f^*(n'))<0$ when $f^*(n')\geq\mu_0^f$. Therefore, $F(\sigma_2)-F(\sigma_1)<0$ is established, and $P(f(n)<f(n')|\sigma)$ decreases as the prediction error $\sigma$ increases when the noise is Gaussian distribution. The above analyses will be added to the revised paper to further elucidate the impact of prediction errors. Under both the uniform error distribution and the Gaussian error distribution, the larger the prediction error, the lower the likelihood of selecting the optimal node.

[1] Xu, Lei, Pingfan Yan, and Tong Chang. "Algorithm cnneim-a and its mean complexity." Proc. of 2nd international conference on computers and applications. IEEE Press, Beijing. 1987.

A2: More intuition on the implications of Theorem 4.3 is provided. In Theorem 4.3, $p_{\sigma}$ is the probability that $f$ of an optimal node $n$ exceeds $f$ of a non-optimal node $n'$,
$p_{\sigma}=P(f(n)\leq f(n')|\sigma)$. $P_S(\sigma)>P_A(\sigma)$ holds if and only if
$$
p_{\sigma}<H(N_o,K),\quad\quad\quad H(N_o,K)=\left(\frac{K}{N_o}\right)^{\frac{1}{N_o-K}}.
$$
$p_{\sigma}$ decreases as the prediction error $\sigma$ increases. If $\sigma$ is quite small, then $p_{\sigma}$ approaches $1$, and the inequality in Theorem 4.3 is unlikely to hold. In this case, A* can identify the optimal solution efficiently without the need for candidate sampling in SeeA*. If $\sigma$ is large, estimated $f$ values are misleading, and the probability that the optimal node's $f$ value is the best among open nodes is low, possiblely even lower than random sampling. In this case, $p_{\sigma}$ is small, and the inequality holds. SeeA* is more effective than A* when $f$ is inaccurate.

As the branching factors increase and the solution paths grow longer, the size of open set $N_o$ grows. $H(N_o,K)$ monotonically increases with respect to $N_o$. As $N_o$ approaches infinity, $H(N_o,K)$ tends to $1$, and the inequality holds. Intuitively, $n$ is expanded if its $f$ value is the smallest among open nodes. Inaccurate predictions raise the likelihood of other nodes having smaller $f$ values. As $N_o$ increases, $f(n)$ is less likely to be the smallest, leading to poorer performance of A*. SeeA* reduces the number of available nodes for selection, resulting in better performance compared to A*.

The number of candidate nodes $K$ is a key hyperparameter balancing exploration and exploitation. In SeeA*, exploitation selects the node with the best $f$ value, like A* search, while exploration uses a sampling strategy to create diverse candidate sets. If $K=1$, the selected node is determined by the sampling strategy, and SeeA* becomes random sampling. If $K\rightarrow\infty$, the candidate set is the same as the open set, and SeeA* degenerates into best-first A*. A smaller $K$ enhances exploration of SeeA*. $H(N_o,K)$ increases with $K$. If $K$ is very small, the value of $H$ will be relatively small. To ensure the inequality holds, an appropriate value of $K$ should be chosen. What's more, the probability of SeeA* expanding the optimal node as defined in Equation 8 reaches its maximum value when $K=-1/\log p_{\sigma}$, which increases with $p_{\sigma}$. For small $p_{\sigma}$, the optimal $K$ is the smallest value 1. When $p_{\sigma}$ approaches 1, the optimal $K$ will be the largest $\infty$. The choice of $K$ value is related to the prediction error $\sigma$ of the heuristic function.

---

### Decision · Program_Chairs · 2024-09-25

**Decision:**

Accept (oral)

**Comment:**

The work builds on  two search algorithms, MCTS an A*, to introduce  a new algorithm, SeeA*, which ehnances A* with a flexible exploration-exploitation balance.  Advantages of SeeA* over A* are justified theoretically and confirmed on real-world and synthetic search domains.

The reviewers seem to agree that the paper brings a significant contribution, is written well, and provides convincing empirical evaluation supported by theoretical analysis. Some reviewers expressed concerns about strong assumptions in the theoretical analysis, but appear to have been convinced by the authors that the theoretical results hold also for more liberal assumption. One reviewer wondered whether the paper fits well within NeurIPS, and the authors justified the appropriateness of the paper for the NeurIPS audience.

The paper introduces an important but simple to grasp enhancement to a broadly used algorithm with many applications in diverse areas of machine learning and artificial intelligence. I recommend the paper for an oral presentation at the conference.